# EMERGENCE OF SUPERPOSITION: UNVEILING THE TRAINING DYNAMICS OF CHAIN OF CONTINUOUS THOUGHT

**Hanlin Zhu**[1]  **Shibo Hao**[2]  **Zhiting Hu**[2]  **Jiantao Jiao**[1,3]  **Stuart Russell**[1]  **Yuandong Tian**[4]
[1] UC Berkeley   [2] UCSD   [3] Nvidia   [4] Meta AI
{hanlinzhu,jiantao,russell}@berkeley.edu
{s5hao,zhh019}@ucsd.edu,  yuandong@meta.com

## ABSTRACT

Previous work shows that the chain of continuous thought (continuous CoT) improves the reasoning capability of large language models (LLMs) by enabling implicit parallel thinking, and a subsequent work provided theoretical insight by showing that a two-layer transformer equipped with continuous CoT can efficiently solve directed graph reachability by maintaining a superposition of multiple reasoning traces in the continuous thought. However, it remains unclear how the superposition mechanism is naturally learned from gradient-based training methods. To fill this gap, we theoretically analyze the training dynamics of a simplified two-layer transformer on the directed graph reachability problem to unveil how the superposition mechanism emerges during training in two training stages – (i) a *thought-generation* stage that autoregressively expands the continuous thought, and (ii) a *prediction* stage that converts the thought into the final answer. Our analysis reveals that during training using continuous thought, the index-matching logit, an important quantity which reflects the strength of the model's local search ability, will first increase and then remain bounded under mild assumptions. The bounded index-matching logit effectively balances exploration and exploitation during the reasoning process: the model will exploit local problem structures to identify plausible search traces, and assign comparable weights to multiple such traces to explore when it is uncertain about which solution is correct, which results in superposition. Our experimental results tracking the growth of logits further validate our theory.

## 1 INTRODUCTIONS

Large Language Models (LLMs) show great reasoning capabilities in many complex tasks, especially when equipped with chain-of-thought (CoT) (Wei et al., 2022). However, due to the large inference cost of long CoT for complex tasks, many recent works seek alternative test-time scaling methods to more efficiently improve LLMs' reasoning ability (Goyal et al., 2023; Wang et al., 2023b; Pfau et al., 2024; Su et al., 2025).

One promising method is to use chain-of-continuous-thought (COCONUT, or continuous CoT) (Hao et al., 2024), where the reasoning trace of an LLM is kept in a continuous latent space instead of projected back to the discrete token space. Continuous CoT exhibits both theoretical advantages (Zhu et al., 2025) and empirical performance gains (Hao et al., 2024) in many tasks. To more efficiently and reliably scale up continuous CoT to solve more challenging tasks, it requires a deeper understanding of its internal mechanism.

Previous work (Zhu et al., 2025) theoretically shows that one of the most important advantages of continuous CoT is that it can enable the model to reason by superposition: when the model encounters multiple plausible search traces and is uncertain about which one is correct, it can keep all plausible traces in parallel since the CoT is in continuous space instead of discrete tokens. In particular, Zhu et al. (2025) abstracted a family of reasoning tasks as a directed graph reachability problem, i.e., whether there exists a path from a given start node to a given destination node, and

showed that a two-layer transformer with $O(n)$ (where $n$ is the number of vertices in the graph) continuous thought decoding steps can efficiently solve the task by providing a construction of the parameters. Therefore, a natural next question is:

> *Do gradient-based methods naturally lead to such a construction, and can we theoretically prove it?*

This paper answers the above question affirmatively by theoretically analyzing the training dynamics of a (simplified) two-layer transformer on the graph reachability problem in two training stages: (i) a *thought generation* stage where the model autoregressively generates a chain of continuous thoughts and (ii) a *prediction* stage where the model predicts the final answer using the generated thought.

Importantly, our analysis of the thought generation training stage reveals why the superposition can *emerge* even if the training data only presents one demonstration for each training sample. Our theoretical analysis as well as experimental results show that when training with continuous thought (i.e., the COCONUT training method in Section 3 and Section 5), the index-matching logit, an important quantity that measures the strength of model's local search capability, will remain bounded under mild conditions, which is in contrast to many previous analysis on transformer logit without continuous thought (e.g., Tian et al. (2023a); Nichani et al. (2024a); Nguyen & Nguyen-Tang (2025)) where the logit will grow logarithmically and thus unbounded. A bounded index-matching logit can balance exploration and exploitation: if the logit is too small, the model cannot even perform local search, resulting in a nearly random guess in the next step; if the logit is too large, the model might over-confidently commit to one of the plausible search traces merely depending on local features (e.g., the indegree of a node in our case) even if it is uncertain about the solution, and thus early discard the correct path. A bounded index-matching logit encourages the model to exploit the local structure while explore multiple plausible solutions by assigning comparable weights to them, which naturally results in superposition. This answers the question of Zhu et al. (2025) why superposition can emerge during training.

## 1.1 RELATED WORKS

**Reasoning with chain of thought.** Chain-of-thought (CoT) (Wei et al., 2022) is a simple yet effective test time scaling method to enhance LLM's reasoning capability. It can either be prompt-based only (Khot et al., 2022; Zhou et al., 2022) or be included in the training sample to create high-quality training data (Yue et al., 2023; Yu et al., 2023; Wang et al., 2023a; Shao et al., 2024). Beyond empirical study, many theoretical works also explore the advantages of the CoT method. For example, Liu et al. (2022); Feng et al. (2023); Merrill & Sabharwal (2023); Li et al. (2024b) shows that CoT can improve the expressivity of transformers. Zhu et al. (2024) studies the importance of CoT for two-hop reasoning via training dynamics. Wen et al. (2024); Kim & Suzuki (2024) studies how CoT in the training data can improve the sample efficiency of transformers. Instead of the text-based CoT, this paper studies continuous CoT where the "thinking tokens" lie in a latent continuous space and do not need to be converted to discrete tokens.

**Latent space reasoning.** A recent line of work studies latent space reasoning, a novel paradigm beyond text-based CoT (Goyal et al., 2023; Wang et al., 2023b; Pfau et al., 2024; Su et al., 2025; Hao et al., 2024). For example, Goyal et al. (2023) proposed to use pause tokens, which are learnable tokens that are inserted into the original text to increase the computation space. Later, London & Kanade (2025) theoretically shows that the pause token can strictly increase the expressivity of the transformer. Similarly, Pfau et al. (2024) studies filler tokens, which also increase the computation space of LLMs. Wang et al. (2023b) proposed to use planning tokens at the beginning of the response generation to improve the reasoning capability. Su et al. (2025) proposed to use abstract tokens in a latent space to enhance the reasoning performance while reducing the inference cost. The most related work is Hao et al. (2024), which proposes to use continuous CoT for reasoning. A follow-up work Zhu et al. (2025) theoretically shows the advantage of continuous CoT via expressivity. Our work takes a further step by analyzing the training dynamics of continuous CoT.

**Training dynamics of transformers.** There is a rich line of literature studying the optimization of transformer-based models (Jelassi et al., 2022; Bietti et al., 2023; Mahankali et al., 2023; Fu et al., 2023; Tian et al., 2023a;b; Zhang et al., 2024; Li et al., 2024a; Huang et al., 2024; Guo et al., 2024).

More recent works also focus on understanding reasoning abilities or patterns of transformers via training dynamics, including induction heads (Nichani et al., 2024a), the reversal curse (Zhu et al., 2024; Ma et al., 2026), CoT (Wen et al., 2024; Kim & Suzuki, 2024; Huang et al., 2025b), factual recall (Nichani et al., 2024b), two hop reasoning (Guo et al., 2025; Lin et al., 2025), out of context reasoning (Huang et al., 2025a), etc. Along the line, our paper aims to understand the internal mechanism of continuous CoT and why superposition emerges via the analysis of training dynamics.

## 2 PROBLEM FORMULATION

**Basic notations.** We use $[N]$ to denote the set $\{1, 2, \ldots, N\}$ for any integer $N > 0$ and use $[i : j]$ to denote $\{i, i+1, \ldots, j-1, j\}$ for integers $i \le j$. For any finite set $\mathcal{X}$, we use $|\mathcal{X}|$ to denote its cardinality and use $\mathsf{Unif}(\mathcal{X})$ to denote the uniform distribution over $\mathcal{X}$. We use $\mathbb{R}$ to denote the set of real numbers and denote $x_+ = \max(x, 0)$ for $x \in \mathbb{R}$. For any vector $\boldsymbol{x} = (x_1, \ldots, x_d) \in \mathbb{R}^d$, the softmax function $\mathsf{SoftMax}(\cdot) : \mathbb{R}^d \to \mathbb{R}^d$ is defined as $\mathsf{SoftMax}(\boldsymbol{x})_i = \exp(x_i)/(\sum_{j=1}^d \exp(x_j))$. Let $\boldsymbol{I}_d \in \mathbb{R}^{d \times d}$ denote the identity matrix. Let $\mathsf{Voc} = [M]$ denote a vocabulary of size $M$ for a fixed integer $M > 0$. For each token $v \in \mathsf{Voc}$, there is an associated embedding $\mathsf{E}(v) \in \mathbb{R}^d$. Let $\mathbf{U} = [\mathsf{E}(1), \mathsf{E}(2), \ldots, \mathsf{E}(M)] \in \mathbb{R}^{d \times M}$ be the token embedding matrix.

**Graph and permutation.** For a directed graph $\mathcal{G} = (\mathcal{V}, \mathcal{E})$ with the vertex set $|\mathcal{V}| = n$ and the edge set $\mathcal{E} = \{(\mathtt{s}_i \to \mathtt{t}_i)\}_{i=1}^m$, we fix a root node $\mathtt{r} \in \mathcal{V}$ and two candidate destination nodes $\mathtt{c}_1, \mathtt{c}_2 \in \mathcal{V}$ such that exactly one node, denoted $\mathtt{c}_\star$, is reachable from $\mathtt{r}$ and denote the other as $\mathtt{c}_\perp$ that is unreachable. For a radius $c \in \mathbb{N}$, define the $c$-ball as $\mathcal{N}_c^{\mathcal{G}}(\mathtt{r}) = \{v \in \mathcal{V} : v \text{ is reachable from } \mathtt{r} \text{ within } c \text{ steps}\}$. For a subset $\mathcal{V}' \subseteq \mathcal{V}$, we define the restricted in-degree as $\deg_{\mathcal{G}, \mathcal{V}'}^-(v) = |\{u \in \mathcal{V}' : (u \to v) \in \mathcal{E}\}|$. We also fix a shortest path from $\mathtt{r}$ to $\mathtt{c}_\star$ as $p = (p_0, \ldots, p_C)$ with $p_0 = \mathtt{r}$, $p_C = \mathtt{c}_\star$, $(p_{c-1} \to p_c) \in \mathcal{E}$ for any $c \in [C]$. For any permutation $\pi$ over $\mathcal{V}$, we define $\pi(\mathcal{G}) = (\mathcal{V}, \pi(\mathcal{E}))$, where $\pi(\mathcal{E}) = \{(\pi(\mathtt{s}) \to \pi(\mathtt{t})) \mid (\mathtt{s} \to \mathtt{t}) \in \mathcal{E}\}$, and define $\pi(p) = (\pi(p_0), \ldots, \pi(p_C))$. We also denote the set of all permutations over $\mathcal{V}$ as $S_{\mathcal{V}}$.

**Chain of continuous thought.** Let $\mathsf{TF}_\theta(\cdot) : (\mathbb{R}^d)^* \to \mathbb{R}^d$ be a transformer which receives an input embedding sequence $\boldsymbol{h} = \boldsymbol{h}_{[t]} \overset{\triangle}{=} (\boldsymbol{h}_1, \boldsymbol{h}_2, \ldots, \boldsymbol{h}_t) \in \mathbb{R}^{d \times t}$ and outputs $\mathsf{TF}_\theta(\boldsymbol{h}) \in \mathbb{R}^d$. For convenience, we assume weight tying. A traditional decoder using a discrete CoT will sample the next token $v_{t+1} \sim \mathsf{SoftMax}(\mathbf{U}^\top \mathsf{TF}_\theta(\boldsymbol{h}))$. Then the embedding of $v_{t+1}$ will be appended to the end of the input, i.e., $\boldsymbol{h}_{t+1} = \mathsf{E}(v_{t+1})$. For continuous CoT, one directly appends the output of the transformer to the end of the input sequence without converting it to a token, i.e., setting $\boldsymbol{h}_{t+1} = \mathsf{TF}_\theta(\boldsymbol{h})$. Assume the prompt $\boldsymbol{x} = [x_1, \ldots, x_{t_0}] \in \mathsf{Voc}^{t_0}$ and its corresponding embedding sequence is $\boldsymbol{h}_{[t_0]} = [\boldsymbol{h}_1, \ldots, \boldsymbol{h}_{t_0}] = [\mathsf{E}(x_1), \ldots, \mathsf{E}(x_{t_0})]$. For notation convenience, we use $[\mathtt{t}_c] = \boldsymbol{h}_{t_0+c}$ to denote the continuous thought generated at decoding step $c$, where $[\mathtt{t}_c] = \mathsf{TF}_\theta(\boldsymbol{h}_{[t_0+c-1]})$. In particular, $[\mathtt{t}_0] = \boldsymbol{h}_{t_0}$. After $C$ decoding steps, one can append a special token $\mathtt{<A>}$ at the end of the sequence to trigger the transformer to switch the mode and generate the final answer. Specifically, one can set $\boldsymbol{h}_T = \mathsf{E}(\mathtt{<A>})$ where $T = t_0 + C + 1$ and generate the final answer $\widetilde{\mathsf{TF}}_{\theta, C, \mathbf{U}}(\boldsymbol{h}_{[t_0]}) := \arg\max_{v \in \mathsf{Voc}} \mathbf{U}^\top \mathsf{TF}_\theta(\boldsymbol{h}_{[T]})$.

**Graph reachability and prompt format.** In this paper, we mainly focus on the directed graph reachability problem as studied in Zhu et al. (2025), where we are given a graph $\mathcal{G} = (\mathcal{V}, \mathcal{E})$, two candidate destination nodes $\mathtt{c}_1$ and $\mathtt{c}_2$, and a root node $\mathtt{r}$. The task is to identify which of the two nodes can be reached by $\mathtt{r}$ (denoted as $\mathtt{c}_\star$). The prompt structure is illustrated in Figure 1 following Zhu et al. (2025). The prompt consists of (1) a BOS (beginning of sentence) token $\mathtt{}$; (2) the graph description part, which contains $m$ edges where each edge is represented by a source node $\mathtt{s}_i$, a target node $\mathtt{t}_i$, and a special edge token $\mathtt{<e>}$; (3) the task description part that contains a special question token $\mathtt{<Q>}$, two candidate destination nodes $\mathtt{c}_1$ and $\mathtt{c}_2$, a special reasoning token $\mathtt{<R>}$ and a root node $\mathtt{r}$. See Table 1 for the full list of token notations. Note that $t_0 = 3m + 6$ is the prompt length, and let $\boldsymbol{h}_{[t_0]} = (\boldsymbol{h}_1, \boldsymbol{h}_2, \ldots \boldsymbol{h}_{t_0})$ be the input embedding sequence. Following Zhu et al. (2025), we use $\mathsf{Idx}(v)$ to denote the position of a token in the input sequence (e.g., $\mathsf{Idx}(\mathtt{}) = 1$, $\mathsf{Idx}(\mathtt{s}_i) = 3i - 1$, $\mathsf{Idx}(\mathtt{c}_1) = 3m + 3$, $\mathsf{Idx}(\mathtt{<R>}) = 3m + 5$), use $\mathsf{Idx}(\mathtt{<e>}, i) = 3i + 1$ to denote the position of the $i$-th $\mathtt{<e>}$ token, and use $\mathsf{Idx}([\mathtt{t}_i]) = t_0 + i$ to denote the position of the continuous thought at step $i$. See Table 2 for the complete list of position indices.

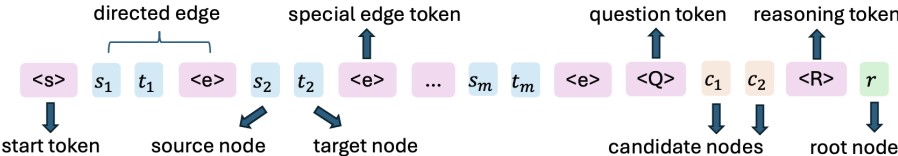

Figure 1: Prompt format of the graph reachability problem (Figure 1 of Zhu et al. (2025)).

Zhu et al. (2025) provided a construction of transformer parameters $\theta$ such that $\widetilde{\mathsf{TF}}_{\theta,C,\mathbf{U}}(\boldsymbol{h}_{[t_0]}) = \mathbf{c}_\star$ (i.e., the transformer can predict the reachable candidate node using continuous CoT) for any graph and root-candidate node tuples, where $\boldsymbol{h}_{[t_0]}$ corresponds to the prompt of the graph and task descriptions. However, they did not theoretically study whether the constructed solution can be naturally learned via gradient-based methods. In the following sections, we theoretically show that the solution can be learned via gradient flow in both the thought generation stage (Section 3) and the prediction stage (Section 4). We also provide empirical results showing that the training dynamics in our theoretical analysis align well with the experiments (Section 5).

## 2.1 INDEX-MATCHING LOGITS AND LOCAL SEARCH CAPABILITY

Before we delve into the technical details in the following sections, we first provide an intuitive explanation of the dynamics of the main mechanism.

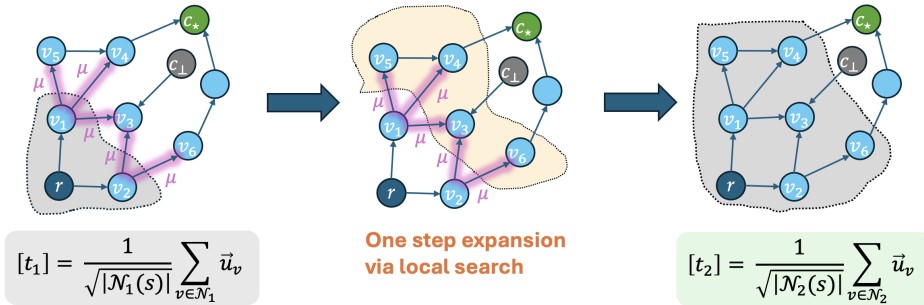

(a) **Left**: The continuous thought at step 1 [t₁] encodes embeddings of nodes that are reachable from the root node r within one step. **Middle**: One-step expansion via local search where the strength is quantified by index-matching logit $\mu$. **Right**: After one-step expansion, the continuous thought at step 2 [t₂] encodes nodes reachable within two steps.

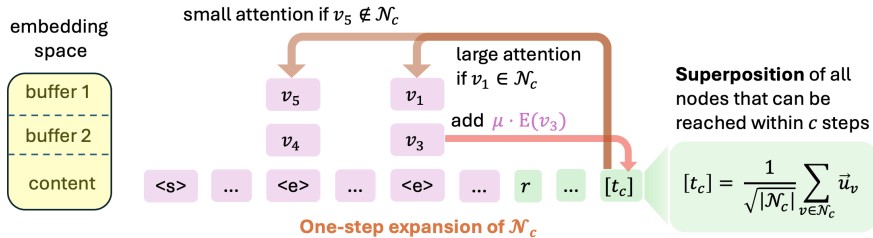

(b) Illustration of how one-step expansion is implemented (adapted from Figure 3 of Zhu et al. (2025)). In the first layer of the transformer, each special edge token <e> copies its corresponding source and target nodes to its buffer spaces. In the second layer, as illustrated in the figure, the current thought [t_c] pays large attention to an edge if its source node has been explored, and adds its target node to the superposition, where the strength of the added node is controlled by the index-matching logit $\mu$. The two edges $v_5 \rightarrow v_4$ and $v_1 \rightarrow v_3$ corresponds to edges in Figure 2a.

Figure 2: Pictorial illustration of the superposition mechanism and the index-matching logit $\mu$.

**Global planning vs. local search.** In the context of graph reachability, the global planning refers to a model's capability to analyze the structure of the whole graph and then determine a path from the root node to the destination node. In contrast, local search focuses only on which nodes are reachable in one step from the current node, which is much easier to learn than global planning. When using discrete CoT, the model can choose only one path at a time. Therefore, the model needs global planning to select the correct path or to backtrack from the wrong one. When using continuous CoT, the model can keep multiple plausible paths simultaneously. Therefore, the model can rely solely on local search to perform parallel BFS, solving the task with only simple skills.

**Index-matching logits.** We use the index-matching logit $\mu$ to quantify the strength of the model's local search capability, which is illustrated in Figure 2 and will be formally defined in (3) in Section 3. In Theorem 1, we will prove that under mild conditions, the index-matching logit $\mu$ will first increase and then remain bounded. Note that a positive, bounded logit $\mu$ effectively balances exploration and exploitation in node expansion: if $\mu$ is too small, each edge will receive similar attention in Figure 2b, and thus the model even lacks the local search capability to exploit the local graph structure; if $\mu$ is too large, the model will put too much weights on nodes with large in-degree (e.g., in Figure 2a, $v_3$ weights $2\mu$ and other frontier nodes such as $v_4$ weights $\mu$, where the difference in weights will be significant under large $\mu$ and commonly used softmax attention) and thus lacks exploration of different plausible paths.

## 3 ANALYSIS OF THE THOUGHT GENERATION STAGE

In this section, we analyze the training dynamics of the thought generation stage. We consider any graph $\mathcal{G} = (\mathcal{V}, \mathcal{E})$, a root node $\mathtt{r} \in \mathcal{V}$, two candidate destination nodes $\mathtt{c}_1, \mathtt{c}_2 \in \mathcal{V}$, where $\{\mathtt{c}_1, \mathtt{c}_2\} \in \{\mathtt{c}_\star, \mathtt{c}_\perp\}$ with $\mathtt{c}_\star$ reachable from $\mathtt{r}$ and $\mathtt{c}_\perp$ unreachable. We are also given a (discrete) CoT demonstration, which is a shortest path $p = (p_0, \ldots, p_C)$ from $\mathtt{r}$ to $\mathtt{c}_\star$ where $p_0 = \mathtt{r}$, $p_C = \mathtt{c}_\star$.

We use curriculum learning following Hao et al. (2024); Zhu et al. (2025), where at stage $(c+1)$ for any $0 \leq c < C$, upon receiving the prompt embeddings $\boldsymbol{h}_{[c_0]}$, the model will first generate $c$ continuous thoughts $[\mathtt{t}_1], \ldots, [\mathtt{t}_c]$ autoregressively without supervision (i.e., no loss calculated on the first $c$ continuous thoughts at stage $c+1$), and then be trained to generate the next continuous thought $[\mathtt{t}_{c+1}] = \mathsf{TF}_\theta(\boldsymbol{h}_{[t_0+c]})$. Since the learning procedure at each stage is similar, we focus below on a fixed $c$.

Zhu et al. (2025) constructs a solution for a two-layer transformer, where the first layer mainly performs copy (e.g., the $i$-th special edge token $<\mathtt{e}>$ will copy the information of its corresponding source node $\mathtt{s}_i$ and target node $\mathtt{t}_i$). Since the copy mechanism has been widely studied (Nguyen & Nguyen-Tang, 2025), as well as its formation via training dynamics (Nichani et al., 2024a), we mainly focus on the dynamics after the copy mechanism has been established. Thus, we analyze the dynamics of the second layer of the transformer.

In particular, let the hidden states of each special edge token $<\mathtt{e}>$ and the current thought $[\mathtt{t}_c]$ after the first transformer layer be

$$\boldsymbol{h}_{\mathsf{Idx}(<\mathtt{e}>,i)} = \mathsf{E}_s(\mathtt{s}_i) + \mathsf{E}_t(\mathtt{t}_i) \in \mathbb{R}^d, \qquad \boldsymbol{h}_{\mathsf{Idx}([\mathtt{t}_c])} = \sum_{v \in \mathcal{N}_c^{\mathcal{G}}(\mathtt{r})} \frac{1}{\sqrt{|\mathcal{N}_c^{\mathcal{G}}(\mathtt{r})|}} \mathsf{E}(v) \in \mathbb{R}^d, \quad (1)$$

where $\mathsf{E}_s(v) \in \mathbb{R}^d$ and $\mathsf{E}_t(v) \in \mathbb{R}^d$ map token $v \in \mathsf{Voc}$ to different subspaces of $\mathbb{R}^d$. For example, as in the construction of Zhu et al. (2025), we can set $d = 3M$, and $\mathsf{E}_s(\cdot)$, $\mathsf{E}_t(\cdot)$ and $\mathsf{E}(\cdot)$ each corresponds to $M$ different non-zero entries. This is also similar to previous work Chen et al. (2025); Nguyen & Nguyen-Tang (2025) where $\mathsf{E}_s(\cdot)$ and $\mathsf{E}_t(\cdot)$ can be viewed as previous token heads. We make the following assumptions on the embedding $\mathsf{E}_s(\cdot)$, $\mathsf{E}_t(\cdot)$ and $\mathsf{E}(\cdot)$:

**Assumption 1** (Orthonormal embeddings). *Assume* $\mathsf{E}_t(\cdot) \equiv \mathsf{E}(\cdot)$. *For any* $u, v \in \mathsf{Voc}$, $\mathsf{E}_s(u)^\top \mathsf{E}_s(v) = \mathsf{E}_t(u)^\top \mathsf{E}_t(v) = \mathbb{1}\{u = v\}$ *and* $\mathsf{E}_s(u)^\top \mathsf{E}_t(v) = 0$.

(1) means after the first layer, each special edge token $<\mathtt{e}>$ will copy the embeddings of its corresponding source and target nodes $\mathtt{s}_i$ and $\mathtt{t}_i$ to the same position in different subspaces. Also, we assume by induction that after training stages $1, 2, \ldots, c$, the current thought generated by the well-trained model $\boldsymbol{h}_{\mathsf{Idx}([\mathtt{t}_c])}$ is a normalized superposition of token embeddings of all nodes reachable from $\mathtt{r}$ within $c$ steps. Below, we study the training dynamics of the current stage (i.e., stage $c+1$).

**The forward path and reparameterization.** We consider the setting where the second layer is attention-only. The forward pass can be formulated as

$$\phi(\boldsymbol{h}; \{\boldsymbol{h}_i\}_i) = \sum_i \mathbf{V}\sigma(\boldsymbol{h}^\top \mathbf{W} \boldsymbol{h}_i)\, \boldsymbol{h}_i,$$

$$\boldsymbol{\xi} = \mathbf{U}^\top\big(\boldsymbol{h}_{\mathsf{Idx}([\mathtt{t_c}])} + \phi(\boldsymbol{h}_{\mathsf{Idx}([\mathtt{t_c}])}; \{\boldsymbol{h}_{\mathsf{Idx}(<\mathtt{e}>,i)}\}_{i=1}^m)\big) \in \mathbb{R}^M,$$

(2)

where $\mathbf{V}, \mathbf{W} \in \mathbb{R}^{d\times d}$ are attention parameters and $\sigma(\cdot) : \mathbb{R} \to \mathbb{R}$ is an activation function that determines the range of attention scores, and $\boldsymbol{\xi} = (\xi_v)_{v\in\mathsf{Voc}} \in \mathbb{R}^M$ is the output logit vector for each token in the vocabulary. Similar to the analysis in Nguyen & Nguyen-Tang (2025), we adopt the linear attention $\sigma(\boldsymbol{h}^\top \mathbf{W}\boldsymbol{h}_i) = \boldsymbol{h}^\top \mathbf{W}\boldsymbol{h}_i$, fix $\mathbf{V} = \boldsymbol{I}$ and use the *index-matching* reparameterization

$$\mathbf{W} = \sum_{v\in\mathcal{V}} \mu_v\, \mathsf{E}(v)\mathsf{E}_{\mathsf{s}}(v)^\top, \quad \mu_v(t) = 0 \text{ for } t = 0. \tag{3}$$

**Remark 1.** *Note that a more general form of the attention weight matrix can be*

$$\mathbf{W} = \mu_{<R>}\, \mathsf{E}(<\mathtt{A}>)\mathsf{E}(<\mathtt{R}>)^\top + \sum_{v,v'\in\mathcal{V}} \mu_{v,v'}\, \mathsf{E}(v)\mathsf{E}_{\mathsf{s}}(v')^\top. \tag{4}$$

*The first term only takes effect in the prediction stage (Section 4), so we can set $\mu_{<R>} = 0$ for now. The second term involves $n \times n$ cross terms. The symmetry of the vertices, which can be enforced by permuting vertex labels during training, makes the $n \times n$ parameters $\{\mu_{v,v'}\}_{v,v'}$ effectively two parameters $\{\mu_1, \mu_2\}$ where $\mu_{v,v} \equiv \mu_1$ and $\mu_{v,v'} \equiv \mu_2$ for $v \neq v'$. Moreover, if we focus on the relative value between $\mu_1$ and $\mu_2$, we can further simplify the attention weight matrix by assuming $\mu_2 = 0$.*

For notation simplicity, we use $\boldsymbol{h}_i$ to denote $\boldsymbol{h}_{\mathsf{Idx}(<\mathtt{e}>,i)}$, use $\boldsymbol{h}_{[\mathtt{t_c}]}$ to denote $\boldsymbol{h}_{\mathsf{Idx}([\mathtt{t_c}])}$ and use $\mathcal{N}_c$, $\mathcal{N}_{c+1}$ to denote $\mathcal{N}_c^{\mathcal{G}}(\mathtt{r})$, $\mathcal{N}_{c+1}^{\mathcal{G}}(\mathtt{r})$, respectively when the graph $\mathcal{G}$ and root node $\mathtt{r}$ is clear from the context. We also denote $d_u := \deg_{\mathcal{G},\mathcal{N}_c}^-(u)$ which is the indegree of $u$ with source nodes restricted in $\mathcal{N}_c$. Finally, we denote $K = |\mathcal{N}_c|$ and $\lambda = \frac{1}{\sqrt{K}}$.

**Loss functions.** An ideal model should be able to directly output the shortest path from the start node $\mathtt{r}$ to the desired candidate destination node $\mathtt{c_\star}$, i.e., the prediction of the $(c+1)$-th continuous thought $[\mathtt{t_{c+1}}]$ exactly corresponds to the $(c+1)$-th step of the shortest path $p_{c+1}$. However, experiments in Zhu et al. (2025) show that even for a 12-layer transformer, it is hard to predict the shortest path even if the length of the shortest path is only 3 or 4. Therefore, we take a step back and pursue a more practical goal – we expect the model to at least be able to generate an arbitrary path starting from the start node $\mathtt{r}$, which only requires local search ability that is much easier than the global planning ability. In the context of continuous thought, we expect the model to include information of all vertices that are reachable from $\mathtt{r}$ within $(c+1)$ steps in the generated thought $[\mathtt{t_{c+1}}]$. We consider the following two loss functions:

$$\text{COCONUT-BFS:} \quad \ell_{\mathcal{G},\mathtt{r}}^{\mathrm{BFS}} := -\log \frac{\sum_{v\in\mathcal{N}_{c+1}} \exp(\xi_v)}{\sum_{v\in\mathcal{V}} \exp(\xi_v)}, \tag{5}$$

$$\text{COCONUT:} \quad \ell_{\mathcal{G},\mathtt{r},p}^{\mathrm{coco}} := -\log \frac{\exp(\xi_{p_{c+1}})}{\sum_{v\in\mathcal{V}} \exp(\xi_v)}, \tag{6}$$

with permutation-averaged dataset losses

$$\mathcal{L}^{\mathrm{BFS}} = \mathbb{E}_{\pi\sim\mathsf{Unif}(S_{\mathcal{V}})}[\ell_{\pi(\mathcal{G}),\pi(\mathtt{r})}^{\mathrm{BFS}}] \quad \text{and} \quad \mathcal{L}^{\mathrm{coco}} = \mathbb{E}_{\pi\sim\mathsf{Unif}(S_{\mathcal{V}})}[\ell_{\pi(\mathcal{G}),\pi(\mathtt{r}),\pi(p)}^{\mathrm{coco}}].$$

Note that, intuitively, the permutation-averaged loss will lead to similar behavior across different parameters. The first loss $\mathcal{L}^{\mathrm{BFS}}$ explicitly encourages the model to predict any nodes in $\mathcal{N}_{C+1}$. However, in practice, it is costly and even impossible to search over the entire solution space exhaustively; instead, we usually present only one demonstration for each task instance during training (in our setting, only one path $p$ per instance $(\mathcal{G}, \mathtt{r}, \mathtt{c_1}, \mathtt{c_2})$), which corresponds to the second loss $\mathcal{L}^{\mathrm{coco}}$ and aligns with the practical setting where chain of thought data can be used for supervision.

Zhu et al. (2025) observed in experiments that superposition emerges even without explicit guidance during training, i.e., using the loss $\mathcal{L}^{\mathrm{coco}}$. In this paper, we investigate the emergence of superposition by analyzing its training dynamics. The following lemma gives the gradient of the index-matching strength parameter $\mu_v(t)$ using gradient flow under the loss function $\mathcal{L}^{\mathrm{coco}}$.

**Lemma 1** (Gradient of $\mu_v$ under $\mathcal{L}^{\text{coco}}$; informal version of Theorem 4 in Appendix B). *Under permutation-averaged training from symmetric initialization and gradient flow $\dot{\mu}_v = -\alpha\nabla_{\mu_v}\mathcal{L}^{\text{coco}}$, we have $\mu_v(t) \equiv \mu(t)$ for all $v$ and times $t$, and the gradient of $\mu_v$ is*

$$\dot{\mu}(t) = \frac{\alpha}{n\sqrt{K}}\Big(d_{p_{c+1}} - F(\mu(t))\Big), \qquad F(\mu) = \frac{\sum_{u\in\mathcal{N}_{c+1}} d_u\, e^{\lambda(\mathbb{1}\{u\in\mathcal{N}_c\}+\mu d_u)}}{\sum_{u\in\mathcal{N}_{c+1}} e^{\lambda(\mathbb{1}\{u\in\mathcal{N}_c\}+\mu d_u)} + (n - |\mathcal{N}_{c+1}|)}.$$

*Moreover $F$ is smooth, strictly increasing, with $F(-\infty) = 0$, $F(+\infty) = \max_{v\in\mathcal{V}} d_v$ and $0 < F(\mu) < \max_{v\in\mathcal{V}} d_v$ for all finite $\mu$.*

The proof is deferred to Appendix B. Note that as long as $d_{p_{c+1}} \neq \max_{v\in\mathcal{V}} d_v$, $\mu(t)$ will converge to $\mu_\star < \infty$. In contrast, under COCONUT-BFS with loss $\mathcal{L}^{\text{BFS}}$, $\mu(t)$ will diverge to infinity. We formalize the comparison into the following theorem and defer the proof to Appendix B.

**Theorem 1** (Bounded vs. divergent attention logits under COCONUT vs. COCONUT-BFS; informal version of Theorem 4 & Lemma 5 in Appendix B). *Let $d_\star := d_{p_{c+1}}$ and $d_{\max} := \max_v d_v$.*

*(i)* *Under COCONUT-BFS (5), $\mu(t)$ grows at least logarithmically in $t$, leading to unbounded attention logits.*

*(ii)* *Under COCONUT (6), if $d_\star < d_{\max}$ then $\mu(t) \to \mu^\star < \infty$, so all attention logits remain uniformly bounded. If $d_\star = d_{\max}$, then $\mu(t) \to \infty$ at least in a logarithmic rate.*

**Emergence of Superposition via Bounded Attention Logits.** By Theorem 1, as long as $F(0) < d_{p_{c+1}} < d_{\max}$, we have $\mu(t) \to \mu^* > 0$. Compared to many previous work (Tian et al., 2023a; Nichani et al., 2024a; Nguyen & Nguyen-Tang, 2025) that analyze the dynamics of attention logits in "discrete" settings where the attention logits diverge to infinity, the COCONUT training method in continuous setting usually result in bounded attention logits. The bounded attention logits lead to a more smooth probability distribution over next tokens, which is beneficial especially under uncertainty: when the model is uncertain about the next step, a more smooth probability distribution under continuous CoT mechanism results in a superposition of different plausible next steps, which implements an effective exploration; on the contrary, an unbounded logit will result in a one-hot-like distribution and thus the model will over-confidently commit to a plausible branch and is likely to discard the ground-truth branch even when the evidence is weak.

Finally, we show that with a positive value of $\mu$, the continuous thought $[\mathtt{t}_{c+1}]$ implements a one-step expansion from $\mathcal{N}_c$ to $\mathcal{N}_{c+1}$ for any graph $\mathcal{G}$ and root node $\mathtt{r}$.

**Theorem 2** (One-step frontier expansion; informal version of Theorem 5 in Appendix B). *For any graph $\mathcal{G}$ and root node $\mathtt{r}$, if the current thought is any positive superposition on $\mathcal{N}_c^{\mathcal{G}}(\mathtt{r})$, i.e., $[\mathtt{t}_c] = \sum_{u\in\mathcal{N}_c} \lambda_u \mathsf{E}(u)$ with $\lambda_u > 0$, then the next thought $[\mathtt{t}_{c+1}]$ satisfies that its token-projected output $\mathbf{U}^\top [\mathtt{t}_{c+1}]$ is supported on the one-step expansion $\mathcal{N}_{c+1}$ and has strictly positive mass on every node in $\mathcal{N}_{c+1}$ if $\mu > 0$. In particular,*

$$\mathbf{U}^\top [\mathtt{t}_{c+1}] = \sum_{v\in\mathcal{N}_{c+1}} \beta_v\, \boldsymbol{e}_v$$

*with*

$$\beta_v = \underbrace{\lambda_v \mathbb{1}\{v\in\mathcal{N}_c\}}_{\text{carryover}} + \mu \underbrace{\sum_{u\in\mathcal{N}_c} \lambda_u \mathbb{1}\{(u\to v)\in\mathcal{E}\}}_{\text{one-hop expansion}} \geq 0.$$

The proof is deferred to Appendix B. Note that at initialization where $\mu = 0$, we have $\beta_v = 0$ for $v \in \mathcal{N}_{c+1}\backslash\mathcal{N}_c$. This means every node outside $\mathcal{N}_c$ has the same attention logits and thus the same next token probability. However, such an exploration is not an effective exploration since it blindly puts the same weight on almost every node in the graph without exploiting the graph structure. Therefore, an appropriate $\mu^* > 0$ effectively balances the exploration and exploitation: (1) it has a positive value so the model can exploit the graph structure and can distinguish nodes within the one-step expansion set; (2) it has a bounded value so it will not overconfidently commit to a plausible branch while discarding other branches merely relying on local structure (such as the indegree of the node) without global planning.

## 4 ANALYSIS OF THE PREDICTION STAGE

In this section, we study how the transformer learns to make the correct prediction $c_\star$ among $\{c_1, c_2\}$ by utilizing the generated continuous thought. Note that according to Section 3, the model is able to generate $[t_C] = \sum_{v \in \mathcal{N}_C} \lambda_v E(v)$ with $\lambda_v \in (0, 1]$, a superposition of all reachable nodes within $C$ steps, via a balanced exploration and exploitation. We denote $\boldsymbol{\lambda} = \{\lambda_v\}_{v \in \mathcal{V}}$. At the final stage, one appends a special answer token `<A>` at the end of the continuous CoT, i.e., $\boldsymbol{h}_T = \boldsymbol{h}_{\texttt{<A>}}$, and make the final prediction $\widetilde{\mathsf{TF}}_{\theta, C, \mathbf{U}}(\boldsymbol{h}_{[t_0]}) := \arg\max_{v \in \mathsf{Voc}} \mathbf{U}^\top \mathsf{TF}_\theta(\boldsymbol{h}_{[T]})$.

**The forward path and reparameterization.** Similar to (2), we formulate the forward pass in the prediction stage as

$$\phi(\boldsymbol{h}; \{\boldsymbol{h}_i\}_i) = \sum_i \mathbf{V}\sigma(\boldsymbol{h}^\top \mathbf{W} \boldsymbol{h}_i)\, \boldsymbol{h}_i,$$

$$\boldsymbol{\xi} = \mathbf{U}^\top\big(\mu_{\texttt{<A>}} \boldsymbol{h}_{\mathsf{Idx}(\texttt{<A>})} + \phi(\boldsymbol{h}_{\mathsf{Idx}(\texttt{<A>})}; \{\boldsymbol{h}_{\texttt{<R>}}\})\big) \in \mathbb{R}^M, \tag{7}$$

where

$$\boldsymbol{h}_{\mathsf{Idx}(\texttt{<R>})} = E(\texttt{<R>}) + E(c_1) + E(c_2), \quad \boldsymbol{h}_{\mathsf{Idx}(\texttt{<A>})} = \boldsymbol{h}_{[t_C]} + E(\texttt{<A>}).$$

Note that after the first transformer layer, the hidden state of `<R>` contains information of two candidate nodes $c_1$ and $c_2$ and the hidden state of `<A>` contains the representation of the last thought $[t_C]$ both due to the copy mechanism in the first layer. Again, we adopt the linear attention $\sigma(\boldsymbol{h}^\top \mathbf{W} \boldsymbol{h}_i) = \boldsymbol{h}^\top \mathbf{W} \boldsymbol{h}_i$, fix $\mathbf{V} = \boldsymbol{I}$ and use the reparameterization

$$\mathbf{W} = \mu_{\texttt{<R>}} E(\texttt{<A>}) E(\texttt{<R>})^\top. \tag{8}$$

**Remark 2.** *The scalar $\mu_{\texttt{<R>}}$ denotes the attention logit strength from `<A>` to `<R>`. The scalar $\mu_{\texttt{<A>}}$ represents the signal strength of the residual stream from the first layer. Also, note that the reparameterization of $\mathbf{W}$ in the prediction stage has a different form from (3) in the thought generation stage. One can either view both (3) and (8) as special cases of a more general version (4) in orthogonal subspaces, or view them as two different attention heads (a thought generation head and a prediction head).*

**The loss function.** In the prediction stage, the goal of the model is to predict the reachable candidate node $c_\star$, and thus the loss function can be written as

$$\ell^{\mathrm{pred}}_{\mathcal{G}, \mathrm{r}, c_1, c_2, \boldsymbol{\lambda}} := -\log \frac{\exp(\xi_{c_\star})}{\sum_{v \in \mathcal{V}} \exp(\xi_v)}, \qquad \mathcal{L}^{\mathrm{pred}} = \mathbb{E}_{(\mathcal{G}, \mathrm{r}, c_1, c_2, \boldsymbol{\lambda}) \sim \mathcal{D}}[\ell^{\mathrm{pred}}_{\mathcal{G}, \mathrm{r}, c_1, c_2, \boldsymbol{\lambda}}], \tag{9}$$

where the $\mathcal{D} = \{(\mathcal{G}^{(i)}, \mathrm{r}^{(i)}, c_1^{(i)}, c_2^{(i)}, \boldsymbol{\lambda}^{(i)})\}_i$ denote the training set. The following lemma provides closed-form logits for each vertex where the proof is deferred to Appendix C.

**Lemma 2** (Closed-form logits; informal version of Lemma 8 in Appendix C). *The logit of each vertex $v \in \mathcal{V}$ has the form*

$$\xi_v = \underbrace{\mu_{\texttt{<A>}} \lambda_v \mathbb{1}\{v \in \mathcal{N}_C\}}_{\textit{residual carryover}} + \underbrace{\mu_{\texttt{<R>}} \mathbb{1}\{v \in \{c_1, c_2\}\}}_{\textit{candidate lift}}.$$

According to Lemma 2, only the candidate node $c_\star$ has both positive residual carryover and candidate lift, and an appropriate relative growth rate of $\mu_{\texttt{<R>}}$ and $\mu_{\texttt{<A>}}$ ensures that $c_\star$ has the largest logit. We formalize the result in the following theorem with proof in Appendix C.

**Theorem 3** (Prediction of the reachable candidate node; informal version of Theorem 6 in Appendix C). *Denote $\mu_A = \mu_{\texttt{<A>}}$ and $\mu_R = \mu_{\texttt{<R>}}$. Let $(\mu_{\texttt{<A>}}(t), \mu_{\texttt{<R>}}(t))$ follow gradient flow on loss defined in (9). Suppose*

$$\lambda_\star := \min_i \lambda_{c_\star}^{(i)} \in (0, 1], \qquad \Delta_{\mathrm{train}} := \max_i \max_{v \in \mathcal{N}_C^{(i)} \setminus \{c_\star^{(i)}\}} \big(\lambda_v^{(i)} - \lambda_{c_\star}^{(i)}\big)_+ \in [0, 1].$$

*Then we have*

$$\frac{(\mu_A(t), \mu_R(t))}{\|(\mu_A(t), \mu_R(t))\|} \to u^\star, \qquad \|(\mu_A(t), \mu_R(t))\| \to \infty,$$

with $u_R^\star/u_A^\star = \lambda_\star + \Delta_{\text{train}}$, and $u_R^\star, u_A^\star > 0$. *Consequently, for any unseen instance* $(\mathcal{G}, r, c_1, c_2, \boldsymbol{\lambda})$ *satisfying* $\lambda_v \in (0,1]$ *on* $\mathcal{N}_C$ *and 0 otherwise, and* $\max_v \lambda_v - \lambda_{c_\star} \leq \Delta_{\text{train}}$, *it holds that:*

$$p_{c_\star}(t) := \frac{\exp\left(\xi_{c_\star}(\mu_A(t), \mu_R(t))\right)}{\sum_v \exp\left(\xi_v(\mu_A(t), \mu_R(t))\right)} \xrightarrow[t\to\infty]{} 1.$$

## 5 EXPERIMENTS

In this section, we present experimental results validating the theoretical analysis. We first describe the setup and overall results, then analyze training dynamics in the thought generation and answer prediction stages.

**Model.** We adopt a GPT-2 style decoder with two transformer layers ($d_{\text{model}}{=}768$, $n_{\text{heads}}{=}8$). The model is trained from scratch with AdamW ($\beta_1{=}0.9$, $\beta_2{=}0.95$, weight decay $10^{-2}$), a constant learning rate of $1 \times 10^{-4}$, and a global batch size of 256.

**Dataset.** We follow the dataset from Zhu et al. (2025), which is a subset of ProsQA (Hao et al., 2024). Different from Zhu et al. (2025), we randomly permute the vertex indices in both training and testing to avoid prediction bias and validate the symmetry assumption in (4). Dataset statistics are summarized in Table 3.

**Training.** Following Hao et al. (2024); Zhu et al. (2025), we use a multi-stage training strategy with supervision from chain-of-thought demonstrations. At stage $c$, the model learns to use $c$ continuous thoughts before predicting the $c$-th node on the reasoning path (*thought-generation* stage). If $c > l$ (the CoT length), the model predicts the final answer after $l$ continuous thoughts and the `<A>` token (*prediction* stage). We train for 150 epochs at Stage 1 and 25 epochs for each subsequent stage, totaling 350 epochs. At each stage, data from earlier stages is mixed in with probability 0.1, which prevents the model from forgetting abilities learned from previous stages. The final accuracy of this model on the test set is 96.2%.

### 5.1 THOUGHT GENERATION

To examine the training dynamics of $\mu_v$ under $\mathcal{L}^{\text{coco}}$, we track the second-layer attention logits. When generating the $c$-th continuous thought, $\mu_v$ corresponds to the logit on an edge token `<e>` whose source lies in $\mathcal{N}_c$. In practice, $\mathcal{L}^{\text{coco}}$ encourages the model to predict the current search frontier rather than revisiting explored nodes, so most attention concentrates on *frontier edges*, i.e., edges with sources in $\mathcal{N}_c \setminus \mathcal{N}_{c-1}$. For theoretical simplicity, we assume $\mu_2 = 0$ in (4). In practice, however, the model does assign non-zero attention logits to other edges. Therefore, we report the logit difference between frontier and non-frontier edges on the test set, which more faithfully reflects the effective value of $\mu_v$.

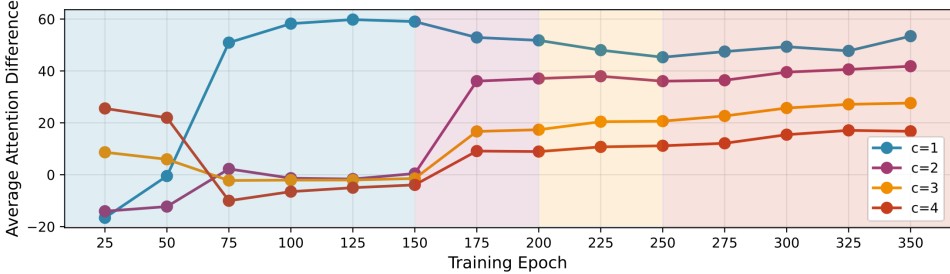

Figure 3: The attention logits difference between frontier edges and others under $\mathcal{L}^{\text{coco}}$ as a proxy for $\mu_v$. The background colors indicate different training stages.

Figure 3 shows the results. In Stage 1 (blue background), the model gradually learns to attend to frontier edges when predicting the first continuous thought ($c = 1$). The logit difference increases

steadily and saturates around 60 after $\sim$125 epochs. This matches the theoretical prediction in Theorem 1: under $\mathcal{L}^{\mathrm{coco}}$, $\mu_v$ first grows and then stabilizes at a bounded value.

When switching to Stage 2 (purple background), the model requires far fewer epochs to establish a positive $\mu$ for $c = 2$. Moreover, this pattern generalizes to $c = 3$ and $c = 4$, even though the model was never explicitly trained to generate more than two continuous thoughts. This "length generalization" indicates that once superposition emerges in earlier stages, later stages can quickly reuse it to expand the frontier further.

We also trained with a variant of $\mathcal{L}^{\mathrm{BFS}}$. Compared to $\mathcal{L}^{\mathrm{coco}}$, the attention logit difference when $c = 1$ did not saturate but kept increasing to much higher values, consistent with the analysis in Theorem 1. Detailed experiments and plots are provided in Appendix E.2.

## 5.2 ANSWER PREDICTION

We next analyze how the model predicts the final answer. According to Lemma 2, the prediction relies on two signals. The first is the *residual carryover*, which brings the explored nodes in the last thought $[\mathtt{t_C}]$ into the answer token with strength $\mu_A$. Concretely, this corresponds to the first-layer attention from $\mathtt{<A>}$ to $[\mathtt{t_C}]$, which copies the superposition of reachable nodes. The second is the *candidate lift*, which raises the logits of the two candidate nodes with strength $\mu_R$. Since $\mathtt{<R>}$ copies the candidate nodes in the first layer, the second-layer attention from $\mathtt{<A>}$ to $\mathtt{<R>}$ serves as a proxy for $\mu_R$.[1]

Figure 4 shows the dynamics of these two proxies. Once training enters the *prediction* stage, both $\mu_A$ and $\mu_R$ increase rapidly and stabilize after roughly 5 epochs. This observation is consistent with Theorem 3, which states that $\mu_A$ and $\mu_R$ grow at comparable rates, ensuring that the reachable candidate $\mathtt{c^\star}$ attains the highest logit. In contrast to the unbounded growth predicted in theory, we observe the logits plateau in practice. A possible reason is that, in practice, prediction-stage training also interacts with thought generation, whereas the theory assumes fixed thought distributions to focus on the relationship between $\mu_R$ and $\mu_A$. We leave a more detailed analysis to future work.

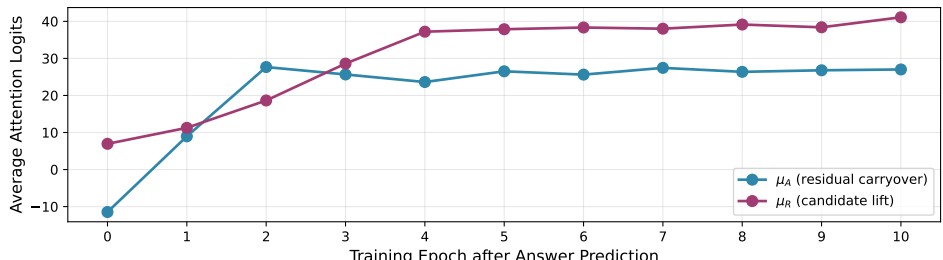

Figure 4: Training dynamics of the proxies for $\mu_A$ (residual carryover) and $\mu_R$ (candidate lift).

## 6 CONCLUSIONS

In this paper, we study the emergence of superposition when training with continuous CoT. In particular, we theoretically analyze the training dynamics of a simplified two-layer transformer on the directed graph reachability problem. Our analysis shows that under mild assumptions, the index-matching logit, an important quantity showing the strength of the model's local search ability, remains bounded during training. A bounded index-matching logit effectively balances exploration and exploitation during the reasoning process and thus enables implicit parallel thinking, which naturally results in superposition. Our experimental results, which track the growth of logits, further validate our theory. We expect our theoretical analysis to bring new insights into a deeper understanding of the mechanism of continuous CoT and ultimately scaling up this promising paradigm more efficiently and reliably.

---

[1]We observe that under different experimental settings and random seeds, the *candidate lift* effect is not always mediated by the $\mathtt{<R>}$ token; alternative attention routes are presented in the Appendix E.3.

ACKNOWLEDGMENTS

This work was partially supported by a gift from Open Philanthropy to the Center for Human-Compatible AI (CHAI) at UC Berkeley, Berkeley Existential Risk Initiative (BERI), and by NSF Grants IIS-1901252 and CCF-2211209. SH is grateful for the support provided by the Bloomberg Data Science Ph.D. Fellowship.

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

# A   NOTATION DETAILS

The notation and meaning of each token and the position index are the same as Zhu et al. (2025). For completeness, we provide detailed descriptions of different tokens in Table 1 (which is Table 2 in Zhu et al. (2025)), and the position index of different tokens or continuous thoughts in Table 2 (which is Table 3 in Zhu et al. (2025)).

| Tokens | Meanings |
|---|---|
| `` | a special token denoting the beginning of the sentence |
| $s_i$ | the source node of edge $i$ |
| $t_i$ | the target node of edge $i$ |
| `<e>` | a special token marking the end of an edge |
| `<Q>` | a special token followed by two candidate nodes |
| $c_1, c_2$ | two candidate destination nodes |
| `<R>` | a special token marking the start of reasoning |
| $r$ | the root node |
| `[t_i]` | the $i$-th continuous thought (represented by a $d$-dimensional vector) |
| `<A>` | a special token driving the model to make the final prediction |

Table 1: Meaning of each token (Table 2 in Zhu et al. (2025)).

| Notations | Position indices |
|---|---|
| $\mathsf{Idx}(\texttt{})$ | $1$ |
| $\mathsf{Idx}(s_i)$ | $3i - 1$ |
| $\mathsf{Idx}(t_i)$ | $3i$ |
| $\mathsf{Idx}(\texttt{<e>}, i)$ | $3i + 1$ |
| $\mathsf{Idx}(\texttt{<Q>})$ | $3m + 2$ |
| $\mathsf{Idx}(c_1)$ | $3m + 3$ |
| $\mathsf{Idx}(c_2)$ | $3m + 4$ |
| $\mathsf{Idx}(\texttt{<R>})$ | $3m + 5$ |
| $\mathsf{Idx}(r)$ | $3m + 6 = t_0$ |
| $\mathsf{Idx}(\texttt{[t_i]})$ | $t_0 + i$ |
| $\mathsf{Idx}(\texttt{<A>})$ | $t_0 + C + 1 = T$ |

Table 2: Position indices of different tokens or continuous thoughts in the input sequence (Table 3 in Zhu et al. (2025)).

# B   MISSING PROOFS FOR SECTION 3

In this section, we provide the full proof of theoretical results in Section 3. We first provide theoretical analysis of COCONUT-BFS and COCONUT in Appendix B.1, Appendix B.2, respectively, and provide results for continuous thought expansion in Appendix B.3.

## B.1   ANALYSIS OF COCONUT-BFS

In this section, we analyze the training dynamics of COCONUT-BFS. We first provide the closed-form formulation of the gradient $\nabla_{\mu_v} \ell^{\mathrm{BFS}}_{\mathcal{G},r} = \nabla_{\mu_v} \ell^{\mathrm{BFS}}_{\mathcal{G},r}(\boldsymbol{\mu})$, where $\boldsymbol{\mu} = \{\mu_v\}_{v \in \mathcal{V}}$ is the set of parameters. We omit the superscript or subscript when the context is clear.

**Lemma 3** (Per-instance gradient of $\mu_v$ for COCONUT-BFS)**.** *Under the loss function of* COCONUT-*BFS as given in* (5) *and the forward pass as in* (2)*, the per-instance gradient is*

$$\nabla_{\mu_v} \ell(\boldsymbol{\mu}) = -\frac{\mathbb{1}\{v \in \mathcal{N}_c\}}{\sqrt{|\mathcal{N}_c|}} \cdot \frac{\sum_{v':(v \to v') \in \mathcal{E}} \exp(\xi_{v'})}{\exp(\xi_+)} \cdot \frac{n - |\mathcal{N}_{c+1}|}{\exp(\xi_+) + n - |\mathcal{N}_{c+1}|}$$

*for any* $v \in \mathcal{V}$*, where* $\xi_+ = \log\left(\sum_{v \in \mathcal{N}_{c+1}} \exp(\xi_v)\right)$*.*

*Proof.* First, note that for any $v \in \mathcal{V}$, according to (2), the logit can be calculated as

$$
\begin{aligned}
\xi_v &= \mathsf{E}(v)^\top \left(\boldsymbol{h}_{[\mathtt{t}_c]} + \phi\left(\boldsymbol{h}_{[\mathtt{t}_c]}; \{\boldsymbol{h}_i\}_{i=1}^m\right)\right) \\
&= \mathsf{E}(v)^\top \left(\boldsymbol{h}_{[\mathtt{t}_c]} + \mathbf{V} \sum_{i=1}^m \sigma\left(\boldsymbol{h}_{[\mathtt{t}_c]}^\top \mathbf{W} \boldsymbol{h}_i\right) \boldsymbol{h}_i\right) \\
&= \mathsf{E}(v)^\top \left(\boldsymbol{h}_{[\mathtt{t}_c]} + \sum_{i=1}^m (\boldsymbol{h}_{[\mathtt{t}_c]}^\top \mathbf{W} \boldsymbol{h}_i) \boldsymbol{h}_i\right) \\
&= \mathsf{E}(v)^\top \left(\boldsymbol{h}_{[\mathtt{t}_c]} + \sum_{i=1}^m \left(\left(\sum_{v' \in \mathcal{N}_c} \lambda \mathsf{E}(v')\right)^\top \sum_{v' \in \mathcal{V}} \mu_{v'} \mathsf{E}(v') \mathsf{E}_\mathsf{s}(v')^\top (\mathsf{E}_\mathsf{s}(\mathsf{s}_i) + \mathsf{E}_\mathsf{t}(\mathtt{t}_i))\right) \boldsymbol{h}_i\right) \\
&= \mathsf{E}(v)^\top \left(\boldsymbol{h}_{[\mathtt{t}_c]} + \sum_{i=1}^m \left(\left(\sum_{v' \in \mathcal{N}_c} \lambda \mathsf{E}(v')\right)^\top \mu_{\mathsf{s}_i} \mathsf{E}(\mathsf{s}_i)\right) \boldsymbol{h}_i\right) \\
&= \mathsf{E}(v)^\top \left(\boldsymbol{h}_{[\mathtt{t}_c]} + \sum_{i=1}^m \lambda \mu_{\mathsf{s}_i} \mathbb{1}\{\mathsf{s}_i \in \mathcal{N}_c\} \boldsymbol{h}_i\right) \\
&= \lambda \cdot \mathbb{1}\{v \in \mathcal{N}_c\} + \sum_{i=1}^m \lambda \mu_{\mathsf{s}_i} \mathbb{1}\{\mathsf{s}_i \in \mathcal{N}_c\} \cdot \mathbb{1}\{v = \mathtt{t}_i\},
\end{aligned}
$$

where $\lambda = \frac{1}{\sqrt{|\mathcal{N}_c|}}$. Note that by the definition of $\mathcal{N}_c$, we have $\xi_v = 0$ if $v \notin \mathcal{N}_{c+1}$. Therefore,

$$
\begin{aligned}
\ell(\boldsymbol{\mu}) &= -\log \frac{\sum_{v \in \mathcal{N}_{c+1}} \exp(\xi_v)}{\sum_{v \in \mathcal{V}} \exp(\xi_v)} \\
&= -\log \frac{\sum_{v \in \mathcal{N}_{c+1}} \exp(\lambda \cdot \mathbb{1}\{v \in \mathcal{N}_c\} + \sum_{i=1}^m \lambda \mu_{\mathsf{s}_i} \mathbb{1}\{\mathsf{s}_i \in \mathcal{N}_c\} \cdot \mathbb{1}\{v = \mathtt{t}_i\})}{\sum_{v \in \mathcal{V}} \exp(\lambda \cdot \mathbb{1}\{v \in \mathcal{N}_c\} + \sum_{i=1}^m \lambda \mu_{\mathsf{s}_i} \mathbb{1}\{\mathsf{s}_i \in \mathcal{N}_c\} \cdot \mathbb{1}\{v = \mathtt{t}_i\})} \\
&= -\log \left(1 - \frac{n - |\mathcal{N}_{c+1}|}{\sum_{v \in \mathcal{V}} \exp(\lambda \cdot \mathbb{1}\{v \in \mathcal{N}_c\} + \sum_{i=1}^m \lambda \mu_{\mathsf{s}_i} \mathbb{1}\{\mathsf{s}_i \in \mathcal{N}_c\} \cdot \mathbb{1}\{v = \mathtt{t}_i\})}\right).
\end{aligned}
$$

For simplicity, we define $\exp(\xi_+) = \sum_{v \in \mathcal{N}_{c+1}} \exp(\xi_v)$ and thus

$$
\ell(\boldsymbol{\mu}) = -\log \left(1 - \frac{n - |\mathcal{N}_{c+1}|}{\exp(\xi_+) + n - |\mathcal{N}_{c+1}|}\right).
$$

Then the per-instance gradient can be calculated as

$$
\begin{aligned}
&\nabla_{\mu_v} \ell(\boldsymbol{\mu}) \\
&= -\frac{\exp(\xi_+) + n - |\mathcal{N}_{c+1}|}{\exp(\xi_+)} \cdot \frac{n - |\mathcal{N}_{c+1}|}{(\exp(\xi_+) + n - |\mathcal{N}_{c+1}|)^2} \cdot \nabla_{\mu_v} \exp(\xi_+) \\
&= -\frac{\sum_{v' \in \mathcal{N}_{c+1}} \exp(\xi_{v'}) \mathbb{1}\{v \in \mathcal{N}_c\} \sum_{i=1}^m \lambda \mathbb{1}\{\mathsf{s}_i = v, \mathtt{t}_i = v'\}}{\exp(\xi_+)} \cdot \frac{n - |\mathcal{N}_{c+1}|}{\exp(\xi_+) + n - |\mathcal{N}_{c+1}|} \\
&= -\lambda \cdot \mathbb{1}\{v \in \mathcal{N}_c\} \frac{\sum_{v':(v \to v') \in \mathcal{E}} \exp(\xi_{v'})}{\exp(\xi_+)} \cdot \frac{n - |\mathcal{N}_{c+1}|}{\exp(\xi_+) + n - |\mathcal{N}_{c+1}|}.
\end{aligned}
$$

$\square$

Now we calculate the gradient of $\mu_v$ over the whole dataset, where the nodes of the graphs are randomly shuffled. We also write $\mathcal{L}^{\mathrm{BFS}} = \mathcal{L}^{\mathrm{BFS}}(\boldsymbol{\mu})$ and omit the superscript when the context is clear.

**Lemma 4** (Whole-dataset gradient of $\mu_v$ for COCONUT-BFS). *Under the loss function of COCONUT-BFS as given in (5) and the forward pass as in (2) and assuming all $\mu_v$ have the same value, the gradient w.r.t. the whole dataset is*

$$
\nabla_{\mu_v} \mathcal{L}(\boldsymbol{\mu}) = -\frac{\exp(-\xi_+)}{n \cdot \sqrt{|\mathcal{N}_c|}} \cdot \frac{n - |\mathcal{N}_{c+1}|}{\exp(\xi_+) + n - |\mathcal{N}_{c+1}|} \sum_{v \in \mathcal{V}} d_v \exp(\xi_v)
$$

*for any $v \in \mathcal{V}$ which is independent of $v$, where $\xi_+ = \log\left(\sum_{v \in \mathcal{N}_{c+1}} \exp\left(\xi_v\right)\right)$.*

*Proof.* Denote $\xi_+^{(\mathcal{G},\mathrm{r})} = \log\left(\sum_{v \in \mathcal{N}_{c+1}^{\mathcal{G}}(\mathrm{r})} \exp\left(\xi_v^{(\mathcal{G},\mathrm{r})}\right)\right)$, where $\xi_v^{(\mathcal{G},\mathrm{r})}$ is the logit of $v$ when the graph in the prompt is $\mathcal{G}$ and the start node is $\mathrm{r}$.

According to Lemma 3 and the condition that all $\mu_v$ have the same value, for any permutation $\pi \in S_{\mathcal{V}}$, we have

$$
\begin{aligned}
&\xi_{\pi(v)}^{(\pi(\mathcal{G}),\pi(\mathrm{r}))} \\
&= \frac{\mathbb{1}\{\pi(v) \in \mathcal{N}_c^{\pi(\mathcal{G})}(\pi(\mathrm{r}))\} + \sum_{i=1}^m \mu_{\pi(\mathrm{s}_i)} \mathbb{1}\{\pi(\mathrm{s}_i) \in \mathcal{N}_c^{\pi(\mathcal{G})}(\pi(\mathrm{r}))\} \cdot \mathbb{1}\{\pi(v) = \pi(\mathrm{t}_i)\}}{\sqrt{\left|\mathcal{N}_c^{\pi(\mathcal{G})}(\pi(\mathrm{r}))\right|}} \\
&= \frac{1}{\sqrt{\left|\mathcal{N}_c^{\mathcal{G}}(\mathrm{r})\right|}} \left( \mathbb{1}\{v \in \mathcal{N}_c^{\mathcal{G}}(\mathrm{r})\} + \sum_{i=1}^m \mu_{\mathrm{s}_i} \mathbb{1}\{\mathrm{s}_i \in \mathcal{N}_c^{\mathcal{G}}(\mathrm{r})\} \cdot \mathbb{1}\{v = \mathrm{t}_i\} \right) \\
&= \xi_v^{(\mathcal{G},\mathrm{r})}.
\end{aligned}
$$

This also implies

$$
\begin{aligned}
\exp\left(\xi_+^{(\pi(\mathcal{G}),\pi(\mathrm{r}))}\right) &= \sum_{\pi(v) \in \mathcal{N}_{c+1}^{\pi(\mathcal{G})}(\pi(\mathrm{r}))} \exp\left(\xi_{\pi(v)}^{(\pi(\mathcal{G}),\pi(\mathrm{r}))}\right) \\
&= \sum_{v \in \mathcal{N}_{c+1}^{\mathcal{G}}(\mathrm{r})} \exp\left(\xi_v^{(\mathcal{G},\mathrm{r})}\right) \\
&= \exp\left(\xi_+^{(\mathcal{G},\mathrm{r})}\right).
\end{aligned}
$$

Therefore, by Lemma 3, we can obtain that

$$
\begin{aligned}
&\nabla_{\mu_{\pi(v)}} \ell_{\pi(\mathcal{G}),\pi(\mathrm{r})}(\boldsymbol{\mu}) \\
&= -\frac{\mathbb{1}\{\pi(v) \in \mathcal{N}_c^{\pi(\mathcal{G})}(\pi(\mathrm{r}))\}}{\sqrt{\left|\mathcal{N}_c^{\pi(\mathcal{G})}(\pi(\mathrm{r}))\right|}} \cdot \frac{\sum_{\pi(v'):(\pi(v) \to \pi(v')) \in \pi(\mathcal{E})} \exp\left(\xi_{\pi(v')}^{(\pi(\mathcal{G}),\pi(\mathrm{r}))}\right)}{\exp\left(\xi_+^{(\pi(\mathcal{G}),\pi(\mathrm{r}))}\right)} \\
&\quad \cdot \frac{n - \left|\mathcal{N}_{c+1}^{\pi(\mathcal{G})}(\pi(\mathrm{r}))\right|}{\exp\left(\xi_+^{(\pi(\mathcal{G}),\pi(\mathrm{r}))}\right) + n - \left|\mathcal{N}_{c+1}^{\pi(\mathcal{G})}(\pi(\mathrm{r}))\right|} \\
&= -\frac{\mathbb{1}\{v \in \mathcal{N}_c^{\mathcal{G}}(\mathrm{r})\}}{\sqrt{\left|\mathcal{N}_c^{\mathcal{G}}(\mathrm{r})\right|}} \cdot \frac{\sum_{v':(v \to v') \in \mathcal{E}} \exp\left(\xi_{v'}^{(\mathcal{G},\mathrm{r})}\right)}{\exp\left(\xi_+^{(\mathcal{G},\mathrm{r})}\right)} \cdot \frac{n - \left|\mathcal{N}_{c+1}^{\mathcal{G}}(\mathrm{r})\right|}{\exp\left(\xi_+^{(\mathcal{G},\mathrm{r})}\right) + n - \left|\mathcal{N}_{c+1}^{\mathcal{G}}(\mathrm{r})\right|} \\
&= \nabla_{\mu_v} \ell_{\mathcal{G},\mathrm{r}}(\boldsymbol{\mu}).
\end{aligned}
$$

Therefore, we can calculate the gradient with respect to the whole dataset as

$$
\begin{aligned}
\nabla_{\mu_v} \mathcal{L}(\boldsymbol{\mu}) &= \mathbb{E}_{\pi \sim \mathsf{Unif}(S_{\mathcal{V}})}[\nabla_{\mu_v} \ell_{\pi(\mathcal{G}),\pi(\mathrm{r})}(\boldsymbol{\mu})] \\
&= \mathbb{E}_{\pi \sim \mathsf{Unif}(S_{\mathcal{V}})}[\nabla_{\mu_{\pi^{-1}(v)}} \ell_{\mathcal{G},\mathrm{r}}(\boldsymbol{\mu})] \\
&= \mathbb{E}_{v' \sim \mathsf{Unif}(\mathcal{V})}[\nabla_{\mu_{v'}} \ell_{\mathcal{G},\mathrm{r}}(\boldsymbol{\mu})]
\end{aligned}
$$

which is independent of $v$ and thus the gradients for $\mu_v$ are equal for all $v \in \mathcal{V}$. Furthermore, we can calculate that

$$\nabla_{\mu_v} \mathcal{L}(\boldsymbol{\mu})$$

$$= \frac{1}{n} \sum_{v \in \mathcal{V}} \left( -\frac{\mathbb{1}\{v \in \mathcal{N}_c\}}{\sqrt{|\mathcal{N}_c|}} \cdot \frac{\sum_{v':(v \to v') \in \mathcal{E}} \exp(\xi_{v'})}{\exp(\xi_+)} \cdot \frac{n - |\mathcal{N}_{c+1}|}{\exp(\xi_+) + n - |\mathcal{N}_{c+1}|} \right)$$

$$= -\frac{1}{n \cdot \sqrt{|\mathcal{N}_c|}} \cdot \frac{n - |\mathcal{N}_{c+1}|}{\exp(\xi_+) + n - |\mathcal{N}_{c+1}|} \sum_{v \in \mathcal{N}_c} \frac{\sum_{v':(v \to v') \in \mathcal{E}} \exp(\xi_{v'})}{\exp(\xi_+)}$$

$$= -\frac{\exp(-\xi_+)}{n \cdot \sqrt{|\mathcal{N}_c|}} \cdot \frac{n - |\mathcal{N}_{c+1}|}{\exp(\xi_+) + n - |\mathcal{N}_{c+1}|} \sum_{v \in \mathcal{N}_c} \sum_{v':(v \to v') \in \mathcal{E}} \exp(\xi_{v'})$$

$$= -\frac{\exp(-\xi_+)}{n \cdot \sqrt{|\mathcal{N}_c|}} \cdot \frac{n - |\mathcal{N}_{c+1}|}{\exp(\xi_+) + n - |\mathcal{N}_{c+1}|} \sum_{v \in \mathcal{V}} d_v \exp(\xi_v).$$

$\square$

According to the gradient of $\mu_v$, we finally show that $\mu_v$ diverges to infinity at least logarithmically in $t$.

**Lemma 5** (Dynamics of $\mu_v$ for COCONUT-BFS). *Let $\mu_v(t)$ be the value of $\mu_v$ at time $t$. Assume zero-initialization, i.e., $\mu_v(0) = 0$ for all $v \in \mathcal{V}$. Under gradient flow*

$$\dot{\mu}_v = -\alpha \cdot \nabla_{\mu_v} \mathcal{L}^{\text{BFS}}(\boldsymbol{\mu}) \tag{10}$$

*where $\alpha > 0$ is the learning rate, we have*

$$\mu_v(t) \geq c_1 \ln(1 + \alpha c_2 t)$$

*for all $v \in \mathcal{V}$ where $c_1 = \frac{1}{2\sqrt{|\mathcal{N}_c|}}$, $c_2 = n^{-3} e^{-2}$.*

*Proof.* First, by Lemma 4, all $\dot{\mu}_v$ have the same value if all $\mu_v$ have the same value. Given that $\mu_v(0) = 0$ for all $v \in \mathcal{V}$, we can obtain that for any time $t$, $\mu_v(t)$ have the same value for all $v \in \mathcal{V}$ using similar argument as in Lemma 15 of Huang et al. (2025a).

Now, given any fixed time $t \geq 0$, it holds that $\mu_v(t)$ has the same value for all $v \in \mathcal{V}$. We omit $t$ for notation convenience, i.e., using $\mu_v$ to represent $\mu_v(t)$. Below, we first provide a lower bound of the gradient $\dot{\mu}_v$.

Since we are guaranteed that one of $\text{c}_1$ and $\text{c}_2$ cannot be reached from $\text{r}$, $\mathcal{N}_{c+1}$ cannot contain all the vertices in $\mathcal{V}$ for any $c$, and thus $n - |\mathcal{N}_{c+1}| \geq 1$. Also, since one of $\text{c}_1$ and $\text{c}_2$ is guaranteed to be reachable from $\text{r}$, there exists $v \in \mathcal{V}$ such that $d_v \geq 1$ for any $c$. This is because the start node $\text{r} \in \mathcal{N}_0 \subseteq \mathcal{N}_c$ for any $c \geq 0$, and we can take $v = p_1$ which is on the shortest path from $\text{r}$ to $\text{c}^\star$. Therefore, we can obtain that $\dot{\mu}_v > 0$. Moreover, we have

$$\dot{\mu}_v = \alpha \cdot \frac{\exp(-\xi_+)}{n \cdot \sqrt{|\mathcal{N}_c|}} \cdot \frac{n - |\mathcal{N}_{c+1}|}{\exp(\xi_+) + n - |\mathcal{N}_{c+1}|} \sum_{v \in \mathcal{V}} d_v \exp(\xi_v)$$

$$\geq \alpha \cdot \frac{\exp(-\xi_+)}{n \cdot \sqrt{|\mathcal{N}_c|}} \cdot \frac{1}{\exp(\xi_+) + 1} \sum_{v \in \mathcal{V}} d_v \exp(\xi_v)$$

$$\geq \frac{\alpha}{n \cdot \sqrt{|\mathcal{N}_c|}} \cdot \frac{\sum_{v \in \mathcal{V}} d_v \exp(\xi_v)}{(\exp(\xi_+) + 1) \cdot \exp(\xi_+)}.$$

Note that by definition, for any vertex $v \in \mathcal{N}_{c+1} \backslash \{\text{r}\}$, there must exists another vertex $v' \in \mathcal{N}_c$ such that $(v' \to v) \in \mathcal{E}$, which implies that $d_v \geq 1$. Therefore,

$$\sum_{v \in \mathcal{V}} d_v \exp(\xi_v) \geq \sum_{v \in \mathcal{N}_{c+1} \backslash \{\text{r}\}} d_v \exp(\xi_v)$$

$$\geq \sum_{v \in \mathcal{N}_{c+1} \backslash \{\text{r}\}} \exp(\xi_v)$$

$$= \exp(\xi_+) - \exp(\xi_\text{r}),$$

which further implies that

$$
\begin{aligned}
\dot{\mu}_v &\geq \frac{\alpha}{n \cdot \sqrt{|\mathcal{N}_c|}} \cdot \frac{\sum_{v \in \mathcal{V}} d_v \exp\left(\xi_v\right)}{\left(\exp\left(\xi_+\right) + 1\right) \cdot \exp\left(\xi_+\right)} \\
&\geq \frac{\alpha}{n \cdot \sqrt{|\mathcal{N}_c|}} \cdot \frac{\exp\left(\xi_+\right) - \exp\left(\xi_r\right)}{\left(\exp\left(\xi_+\right) + 1\right) \cdot \exp\left(\xi_+\right)}.
\end{aligned}
$$

Now recall from Lemma 3 that

$$
\begin{aligned}
\xi_v &= \frac{1}{\sqrt{|\mathcal{N}_c|}} \left( \mathbb{1}\{v \in \mathcal{N}_c\} + \sum_{i=1}^{m} \mu_{\mathsf{s}_i} \mathbb{1}\{\mathsf{s}_i \in \mathcal{N}_c\} \cdot \mathbb{1}\{v = \mathsf{t}_i\} \right) \\
&= \frac{1}{\sqrt{|\mathcal{N}_c|}} \left( \mathbb{1}\{v \in \mathcal{N}_c\} + d_v \cdot \mu_v \right) \\
&\leq \frac{1}{\sqrt{|\mathcal{N}_c|}} \left( 1 + |\mathcal{N}_c| \cdot \mu_v \right).
\end{aligned}
$$

Therefore,

$$
\begin{aligned}
\exp\left(\xi_+\right) &= \sum_{v \in \mathcal{N}_{c+1}} \exp\left(\xi_v\right) \\
&\leq |\mathcal{N}_{c+1}| \exp\left( \frac{1}{\sqrt{|\mathcal{N}_c|}} \left(1 + |\mathcal{N}_c| \cdot \mu_v\right) \right) \\
&\leq n \cdot \exp\left( 1 + \sqrt{|\mathcal{N}_c|} \cdot \mu_v \right).
\end{aligned}
$$

Also, since $p_1 \in \mathcal{N}_{c+1}$ and $\deg_{\mathcal{G}, \mathcal{N}_c}^{-}(p_1) \geq 1$, we can obtain that

$$
\begin{aligned}
\exp\left(\xi_+\right) - \exp\left(\xi_r\right) &\geq \exp\left( \xi_{p_1}^{(\mathcal{G}, r)} \right) \\
&= \exp\left( \frac{1}{\sqrt{|\mathcal{N}_c|}} \left( \mathbb{1}\{p_1 \in \mathcal{N}_c\} + d_{p_1} \cdot \mu_v \right) \right) \\
&\geq \exp\left( \frac{\mu_v}{\sqrt{|\mathcal{N}_c|}} \right).
\end{aligned}
$$

Combining the above two inequalities, we can obtain that

$$
\begin{aligned}
\dot{\mu}_v &\geq \frac{\alpha}{n \cdot \sqrt{|\mathcal{N}_c|}} \cdot \frac{\exp\left(\xi_+\right) - \exp\left(\xi_r\right)}{\left(\exp\left(\xi_+\right) + 1\right) \cdot \exp\left(\xi_+\right)} \\
&\geq \frac{\alpha}{n \cdot \sqrt{|\mathcal{N}_c|}} \cdot \frac{\exp\left( \frac{\mu_v}{\sqrt{|\mathcal{N}_c|}} \right)}{2n^2 \cdot \exp\left( 2 + 2\sqrt{|\mathcal{N}_c|} \cdot \mu_v \right)} \\
&\geq \frac{\alpha}{2n^3 e^2 \cdot \sqrt{|\mathcal{N}_c|}} \cdot \exp\left( -2\sqrt{|\mathcal{N}_c|} \cdot \mu_v \right).
\end{aligned}
$$

Finally, by applying Lemma 13, we can obtain that

$$
\mu_v(t) \geq \frac{1}{2\sqrt{|\mathcal{N}_c|}} \ln\left( 1 + \alpha n^{-3} e^{-2} t \right).
$$

$\square$

## B.2 ANALYSIS OF COCONUT

In this section, we analyze the training dynamics of COCONUT. Similarly, we first provide the closed-form formulation of the gradient $\nabla_{\mu_v} \ell_{\mathcal{G}, r, p}^{\text{coco}} = \nabla_{\mu_v} \ell_{\mathcal{G}, r, p}^{\text{coco}}(\boldsymbol{\mu})$, where $\boldsymbol{\mu} = \{\mu_v\}_{v \in \mathcal{V}}$ is the set of parameters. We omit the superscript or subscript when the context is clear.

**Lemma 6** (Per-instance gradient of $\mu_v$ for COCONUT). *Under the loss function of* COCONUT *as given in* (6) *and the forward pass as in* (2)*, the per-instance gradient is*

$$\nabla_{\mu_v}\ell(\boldsymbol{\mu}) = \frac{\mathbb{1}\{v \in \mathcal{N}_c\}}{\sqrt{|\mathcal{N}_c|}}\left(-\mathbb{1}\{(v \to p_{c+1}) \in \mathcal{E}\} + \frac{\sum_{v':(v \to v') \in \mathcal{E}} \exp\left(\xi_{v'}\right)}{\exp\left(\xi_+\right) + n - |\mathcal{N}_{c+1}|}\right)$$

*for any $v \in \mathcal{V}$, where $\xi_+ = \log\left(\sum_{v \in \mathcal{N}_{c+1}} \exp\left(\xi_v\right)\right)$.*

*Proof.* First, according to the proof of Lemma 3, we have

$$\xi_v = \lambda \cdot \mathbb{1}\{v \in \mathcal{N}_c\} + \sum_{i=1}^{m} \lambda \mu_{\mathsf{s}_i} \mathbb{1}\{\mathsf{s}_i \in \mathcal{N}_c\} \cdot \mathbb{1}\{v = \mathsf{t}_i\},$$

where $\lambda = \frac{1}{\sqrt{|\mathcal{N}_c|}}$. Note that by the definition of $\mathcal{N}_c$, we have $\xi_v = 0$ if $v \notin \mathcal{N}_{c+1}$. Therefore,

$$\ell(\boldsymbol{\mu}) = -\log\frac{\exp\left(\xi_{p_{c+1}}\right)}{\sum_{v \in \mathcal{V}} \exp\left(\xi_v\right)} = -\log\left(\frac{\exp\left(\xi_{p_{c+1}}\right)}{\exp\left(\xi_+\right) + n - |\mathcal{N}_{c+1}|}\right),$$

where $\exp\left(\xi_+\right) = \sum_{v \in \mathcal{N}_{c+1}} \exp\left(\xi_v\right)$.

Then the per-instance gradient can be calculated as

$$\nabla_{\mu_v}\ell(\boldsymbol{\mu})$$
$$= -\frac{\exp\left(\xi_+\right) + n - |\mathcal{N}_{c+1}|}{\exp\left(\xi_{p_{c+1}}\right)}$$
$$\cdot \frac{\nabla_{\mu_v} \exp\left(\xi_{p_{c+1}}\right) \cdot \left(\exp\left(\xi_+\right) + n - |\mathcal{N}_{c+1}|\right) - \exp\left(\xi_{p_{c+1}}\right) \nabla_{\mu_v} \exp\left(\xi_+\right)}{\left(\exp\left(\xi_+\right) + n - |\mathcal{N}_{c+1}|\right)^2}$$
$$= -\frac{\nabla_{\mu_v} \exp\left(\xi_{p_{c+1}}\right)}{\exp\left(\xi_{p_{c+1}}\right)} + \frac{\nabla_{\mu_v} \exp\left(\xi_+\right)}{\exp\left(\xi_+\right) + n - |\mathcal{N}_{c+1}|}$$

$$= -\nabla_{\mu_v}\xi_{p_{c+1}} + \frac{\nabla_{\mu_v} \exp\left(\xi_+\right)}{\exp\left(\xi_+\right) + n - |\mathcal{N}_{c+1}|}.$$

Since

$$\nabla_{\mu_v} \exp\left(\xi_+\right) = \sum_{v' \in \mathcal{N}_{c+1}} \exp\left(\xi_{v'}\right) \mathbb{1}\{v \in \mathcal{N}_c\}\sum_{i=1}^{m} \lambda \mathbb{1}\{\mathsf{s}_i = v, \mathsf{t}_i = v'\}$$
$$= \lambda \cdot \mathbb{1}\{v \in \mathcal{N}_c\}\sum_{v':(v \to v') \in \mathcal{E}} \exp\left(\xi_{v'}\right)$$

and

$$\nabla_{\mu_v}\xi_{p_{c+1}} = \lambda \cdot \mathbb{1}\{v \in \mathcal{N}_c\} \cdot \mathbb{1}\{(v \to p_{c+1}) \in \mathcal{E}\},$$

we can finally obtain that

$$\nabla_{\mu_v}\ell(\boldsymbol{\mu}) = \frac{\mathbb{1}\{v \in \mathcal{N}_c\}}{\sqrt{|\mathcal{N}_c|}}\left(-\mathbb{1}\{(v \to p_{c+1}) \in \mathcal{E}\} + \frac{\sum_{v':(v \to v') \in \mathcal{E}} \exp\left(\xi_{v'}\right)}{\exp\left(\xi_+\right) + n - |\mathcal{N}_{c+1}|}\right).$$

$\square$

Now we calculate the gradient of $\mu_v$ over the whole dataset, where the nodes of the graphs are randomly shuffled. We also write $\mathcal{L}^{\text{coco}} = \mathcal{L}^{\text{coco}}(\boldsymbol{\mu})$ and omit the superscript when the context is clear.

**Lemma 7** (Whole-dataset gradient of $\mu_v$ for COCONUT). *Under the loss function of COCONUT as given in (6) and the forward pass as in (2) and assuming all $\mu_v$ have the same value, the gradient w.r.t. the whole dataset is*

$$\nabla_{\mu_v}\mathcal{L}(\boldsymbol{\mu}) = \frac{1}{n \cdot \sqrt{|\mathcal{N}_c|}}\left(-d_{p_{c+1}} + \frac{\sum_{v \in \mathcal{N}_{c+1}} d_v \exp\left(\xi_v\right)}{\exp\left(\xi_+\right) + n - |\mathcal{N}_{c+1}|}\right)$$

*for any $v \in \mathcal{V}$ which is independent of $v$, where $\xi_+ = \log\left(\sum_{v \in \mathcal{N}_{c+1}} \exp\left(\xi_v\right)\right)$.*

*Proof.* Similar to Lemma 4, we denote $\xi_+^{(\mathcal{G},\mathbf{r})} = \log\left(\sum_{v \in \mathcal{N}_{c+1}^{\mathcal{G}}(\mathbf{r})} \exp\left(\xi_v^{(\mathcal{G},\mathbf{r})}\right)\right)$, where $\xi_v^{(\mathcal{G},\mathbf{r})}$ is the logit of $v$ when the graph in the prompt is $\mathcal{G}$ and the start node is $\mathbf{r}$. According to the proof of Lemma 4, for any permutation $\pi \in S_{\mathcal{V}}$, we have

$$\xi_{\pi(v)}^{(\pi(\mathcal{G}),\pi(\mathbf{r}))} = \xi_v^{(\mathcal{G},\mathbf{r})}$$

for any $v \in \mathcal{V}$ and

$$\exp\left(\xi_+^{(\pi(\mathcal{G}),\pi(\mathbf{r}))}\right) = \exp\left(\xi_+^{(\mathcal{G},\mathbf{r})}\right).$$

Therefore, by Lemma 6, we can obtain that

$$\nabla_{\mu_{\pi(v)}} \ell_{\pi(\mathcal{G}),\pi(\mathbf{r}),\pi(p)}(\boldsymbol{\mu})$$
$$=\frac{\mathbb{1}\{\pi(v) \in \mathcal{N}_c^{\pi(\mathcal{G})}(\pi(\mathbf{r}))\}}{\sqrt{\left|\mathcal{N}_c^{\pi(\mathcal{G})}(\pi(\mathbf{r}))\right|}}\left(-\mathbb{1}\{(\pi(v) \to \pi(p_{c+1})) \in \pi(\mathcal{E})\}\right.$$
$$\left.+ \frac{\sum_{\pi(v'):(\pi(v) \to \pi(v')) \in \pi(\mathcal{E})} \exp\left(\xi_{\pi(v')}^{(\pi(\mathcal{G}),\pi(\mathbf{r}))}\right)}{\exp\left(\xi_+^{(\pi(\mathcal{G}),\pi(\mathbf{r}))}\right) + n - \left|\mathcal{N}_{c+1}^{\pi(\mathcal{G})}(\pi(\mathbf{r}))\right|}\right)$$
$$=\frac{\mathbb{1}\{v \in \mathcal{N}_c^{\mathcal{G}}(\mathbf{r})\}}{\sqrt{|\mathcal{N}_c^{\mathcal{G}}(\mathbf{r})|}}\left(-\mathbb{1}\{(v \to p_{c+1}) \in \mathcal{E}\} + \frac{\sum_{v':(v \to v') \in \mathcal{E}} \exp\left(\xi_{v'}^{(\mathcal{G},\mathbf{r})}\right)}{\exp\left(\xi_+^{(\mathcal{G},\mathbf{r})}\right) + n - |\mathcal{N}_{c+1}^{\mathcal{G}}(\mathbf{r})|}\right)$$
$$=\nabla_{\mu_v}\ell_{\mathcal{G},\mathbf{r},p}(\boldsymbol{\mu}).$$

Therefore, we can calculate the gradient with respect to the whole dataset as

$$\nabla_{\mu_v}\mathcal{L}(\boldsymbol{\mu}) = \mathbb{E}_{\pi \sim \mathsf{Unif}(S_{\mathcal{V}})}[\nabla_{\mu_v}\ell_{\pi(\mathcal{G}),\pi(\mathbf{r}),\pi(p)}(\boldsymbol{\mu})]$$
$$= \mathbb{E}_{\pi \sim \mathsf{Unif}(S_{\mathcal{V}})}[\nabla_{\mu_{\pi^{-1}(v)}}\ell_{\mathcal{G},\mathbf{r},p,c}(\boldsymbol{\mu})]$$
$$= \mathbb{E}_{v' \sim \mathsf{Unif}(\mathcal{V})}[\nabla_{\mu_{v'}}\ell_{\mathcal{G},\mathbf{r},p,c}(\boldsymbol{\mu})]$$

which is independent of $v$ and thus the gradients for $\mu_v$ are equal for all $v \in \mathcal{V}$. Furthermore, similar to Lemma 4, we can calculate that

$$\nabla_{\mu_v}\mathcal{L}(\boldsymbol{\mu})$$
$$=\frac{1}{n}\sum_{v \in \mathcal{V}}\frac{\mathbb{1}\{v \in \mathcal{N}_c\}}{\sqrt{|\mathcal{N}_c|}}\left(-\mathbb{1}\{(v \to p_{c+1}) \in \mathcal{E}\} + \frac{\sum_{v':(v \to v') \in \mathcal{E}} \exp\left(\xi_{v'}\right)}{\exp\left(\xi_+^{(\mathcal{G},\mathbf{r})}\right) + n - |\mathcal{N}_{c+1}|}\right)$$
$$=\frac{1}{n \cdot \sqrt{|\mathcal{N}_c|}}\left(-d_{p_{c+1}} + \frac{\sum_{v \in \mathcal{N}_{c+1}} d_v \exp\left(\xi_v\right)}{\exp\left(\xi_+^{(\mathcal{G},\mathbf{r})}\right) + n - |\mathcal{N}_{c+1}|}\right).$$

$\square$

Finally, we derive the dynamics of $\mu_v$. Recall that we denote $K = |\mathcal{N}_c|, \lambda = \frac{1}{\sqrt{K}}$. We also make the following notation for Theorem 4. Let $d_\star := d_{p_{c+1}}$ and $d_{\max} := \max_{u \in \mathcal{V}} d_u$. Moreover, Let

$c_0 := n - |\mathcal{N}_{c+1}| \geq 1$ and denote

$$\xi_u(\mu) := \lambda\left(\mathbb{1}\{u \in \mathcal{N}_c\} + \mu\, d_u\right), \qquad E_+(\mu) := \sum_{u \in \mathcal{N}_{c+1}} e^{\xi_u(\mu)},$$

$$S(\mu) := \sum_{u \in \mathcal{N}_{c+1}} d_u\, e^{\xi_u(\mu)}, \qquad F(\mu) := \frac{S(\mu)}{E_+(\mu) + c_0}.$$

**Theorem 4** (Dynamics of $\mu_v$ for COCONUT). *Let $\mu_v(t)$ be the value of $\mu_v$ at time t. Assume zero-initialization, i.e., $\mu_v(0) = 0$ for all $v \in \mathcal{V}$. Under gradient flow*

$$\dot{\mu}_v = -\alpha \cdot \nabla_{\mu_v} \mathcal{L}^{\text{coco}}(\boldsymbol{\mu}) \tag{11}$$

*where $\alpha > 0$ is the learning rate, suppose the initialization satisfies $\mu_v(0) = 0$ for all $v$. Then:*

1. ***Scalar reduction.*** *For all $t \geq 0$, $\mu_v(t) \equiv \mu(t)$ is shared across $v$, and $\mu(t)$ satisfies*

$$\boxed{\dot{\mu}(t) = \frac{\alpha}{n\sqrt{K}}\left(d_\star - F(\mu(t))\right).} \tag{12}$$

2. ***Regularity of*** *$F$. The function $F : \mathbb{R} \to \mathbb{R}$ is $C^\infty$, strictly increasing, and satisfies*

$$\lim_{\mu \to -\infty} F(\mu) = 0, \qquad \lim_{\mu \to +\infty} F(\mu) = d_{\max}, \qquad 0 < F(\mu) < d_{\max} \quad \text{for all finite } \mu.$$

3. ***Finite fixed point when*** *$d_\star < d_{\max}$. If $d_\star < d_{\max}$, there exists a unique $\mu^\star \in \mathbb{R}$ such that $F(\mu^\star) = d_\star$. The solution $\mu(t)$ of (12) with $\mu(0) = 0$ converges monotonically to $\mu^\star$:*

$$\mu(t) \nearrow \mu^\star \quad \text{if } F(0) \leq d_\star, \qquad \mu(t) \searrow \mu^\star \quad \text{if } F(0) > d_\star,$$

*and the equilibrium $\mu^\star$ is locally exponentially stable, i.e., there exists $\gamma > 0$ such that for all large enought $t$, it holds that*

$$|\mu(t) - \mu^\star| \leq e^{-\gamma t}\, |\mu(0) - \mu^\star|.$$

4. ***Logarithmic divergence when*** *$d_\star = d_{\max}$. If $d_\star = d_{\max}$, then $\dot{\mu}(t) > 0$ for all $t$ and $\mu(t) \to +\infty$. Moreover, for all $t \geq 0$,*

$$\boxed{\mu(t) \geq \frac{1}{\lambda d_{\max}} \ln\left(1 + \frac{\alpha\, \lambda\, d_{\max}^2\, c_0\, e^{-\lambda}}{2\, n^2 \sqrt{K}}\, t\right).} \tag{13}$$

*Proof.* **(1) Scalar reduction.** By Lemma 7 and the similar argument as in the proof of Lemma 5, we have $\mu_v(t) \equiv \mu(t)$ for all $t \geq 0$. Therefore, the gradient $\nabla_{\mu_v} \mathcal{L}(\boldsymbol{\mu})$ is independent of $v$ and equals

$$\nabla_{\mu_v} \mathcal{L}(\boldsymbol{\mu}) = \frac{1}{n\sqrt{K}}\left(-d_\star + \frac{\sum_{u \in \mathcal{N}_{c+1}} d_u\, e^{\xi_u(\mu_u)}}{\sum_{u \in \mathcal{N}_{c+1}} e^{\xi_u(\mu_u)} + n - |\mathcal{N}_{c+1}|}\right) = \frac{1}{n\sqrt{K}}\left(-d_\star + F(\mu_v)\right).$$

Thus, we have

$$\dot{\mu}(t) = -\nabla_{\mu_v} \mathcal{L}(\boldsymbol{\mu}(t)) = \frac{\alpha}{n\sqrt{K}}\left(d_\star - F(\mu(t))\right).$$

**(2) Regularity and limits of** *$F$.* By the proof of Lemma 6 and the condition that $\mu_v(t) \equiv \mu(t)$ for all $v \in \mathcal{V}$ and $t \geq 0$, we have

$$E_+(\mu) = \sum_{u \in \mathcal{N}_{c+1}} \exp(\lambda(\mathbb{1}\{u \in \mathcal{N}_c\} + \mu d_u)), S(\mu) = \sum_{u \in \mathcal{N}_{c+1}} d_u \exp(\lambda(\mathbb{1}\{u \in \mathcal{N}_c\} + \mu d_u)).$$

Both functions are finite sums of $C^\infty$ functions of $\mu$, hence $F(\mu) = S(\mu)/(E_+(\mu) + c_0)$ is also $C^\infty$.

Now we show the strict monotonicity of $F(\cdot)$ on $\mu$ by differentiation. We further write $\xi_u := \xi_u(\mu)$ for brevity. Then

$$E'_+(\mu) = \lambda \sum_{u \in \mathcal{N}_{c+1}} d_u e^{\xi_u}, \qquad S'(\mu) = \lambda \sum_{u \in \mathcal{N}_{c+1}} d_u^2 e^{\xi_u}.$$

Therefore, we can obtain that

$$F'(\mu) = \frac{S'(\mu)\,(E_+(\mu) + c_0) - S(\mu)E'_+(\mu)}{(E_+(\mu) + c_0)^2}$$

$$= \frac{\lambda}{(E_+(\mu) + c_0)^2} \left[ \left( \sum_{u \in \mathcal{N}_{c+1}} d_u^2 e^{\xi_u} \right) (E_+(\mu) + c_0) - \left( \sum_{u \in \mathcal{N}_{c+1}} d_u e^{\xi_u} \right)^2 \right].$$

Note that by the Cauchy-Schwarz inequality, we have

$$\sum_{u \in \mathcal{N}_{c+1}} d_u^2 e^{\xi_u} \cdot E_+(\mu) - \left( \sum_{u \in \mathcal{N}_{c+1}} d_u e^{\xi_u} \right)^2 \geq 0,$$

and hence

$$F'(\mu) \geq \frac{\lambda}{(E_+(\mu) + c_0)^2} c_0 \, E_+(\mu) \left( \sum_{u \in \mathcal{N}_{c+1}} d_u^2 e^{\xi_u} \right) > 0$$

since $c_0 > 0$ and there exists at least one node $u \in \mathcal{N}_{c+1}$ (e.g., $p_{c+1}$) such that $d_u \geq 1$ by definition. Thus, $F(\cdot)$ is strictly increasing.

Now we consider the limits of $F(\cdot)$. First, note that

$$S(\mu) = \sum_{u \in \mathcal{N}_{c+1}} d_u \, e^{\xi_u(\mu)},$$

and for each $u \in \mathcal{N}_{c+1}$, either $d_u = 0$ or $d_u > 0$ and thus

$$\lim_{\mu \to -\infty} d_u e^{\xi_u(\mu)} = \lim_{\mu \to -\infty} d_u \exp(\lambda(\mathbb{1}\{u \in \mathcal{N}_c\} + \mu d_u)) = 0.$$

Therefore, we have $\lim_{\mu \to -\infty} S(\mu) = 0$. Moreover, since $E_+(\mu) + c_0 \geq c_0 > 0$, we have

$$\lim_{\mu \to -\infty} F(\mu) = 0.$$

Now we consider the case when $\mu \to +\infty$. Since

$$\frac{e^{\xi_u(\mu)}}{E_+(\mu) + c_0} = \frac{\exp\left(\lambda(\mathbb{1}\{u \in \mathcal{N}_c\} + \mu d_u)\right)}{\sum_{v \in \mathcal{N}_{c+1}} \exp(\lambda(\mathbb{1}\{v \in \mathcal{N}_c\} + \mu d_v)) + c_0}$$

$$= \frac{1}{\sum_{v \in \mathcal{N}_{c+1}} \exp(\lambda(\mathbb{1}\{v \in \mathcal{N}_c\} - \mathbb{1}\{u \in \mathcal{N}_c\} + \mu(d_v - d_u))) + c_0}.$$

As $\mu \to +\infty$, we can obtain that if $d_u < d_{\max}$, then

$$\lim_{\mu \to +\infty} \frac{e^{\xi_u(\mu)}}{E_+(\mu) + c_0} \leq \lim_{\mu \to +\infty} \frac{1}{\exp(\lambda(-1 + \mu(d_{\max} - d_u))) + c_0} = 0.$$

If $d_u = d_{\max}$, then

$$\lim_{\mu \to +\infty} \frac{e^{\xi_u(\mu)}}{E_+(\mu) + c_0}$$

$$= \lim_{\mu \to +\infty} \frac{\exp\left(\lambda \cdot \mathbb{1}\{u \in \mathcal{N}_c\}\right)}{\sum_{v \in D_{\max}} \exp(\lambda \cdot \mathbb{1}\{v \in \mathcal{N}_c\}) + \sum_{v \in \mathcal{N}_{c+1} \setminus D_{\max}} \exp(\lambda(\mathbb{1}\{v \in \mathcal{N}_c\} + \mu(d_v - d_{\max}))) + \frac{c_0}{e^{\lambda \mu d_{\max}}}}$$

$$= \frac{\exp\left(\lambda \cdot \mathbb{1}\{u \in \mathcal{N}_c\}\right)}{\sum_{v \in D_{\max}} \exp(\lambda \cdot \mathbb{1}\{v \in \mathcal{N}_c\})},$$

where $D_{\max} := \{u \in \mathcal{N}_{c+1} : d_u = d_{\max}\}$. Therefore,

$$\lim_{\mu \to +\infty} F(\mu) = \sum_{u \in D_{\max}} d_u \frac{\exp\left(\lambda \cdot \mathbb{1}\{u \in \mathcal{N}_c\}\right)}{\sum_{v \in D_{\max}} \exp(\lambda \cdot \mathbb{1}\{v \in \mathcal{N}_c\})} = d_{\max}.$$

Finally, the inequality $F(\mu) < d_{\max}$ for finite $\mu$ follows from $S(\mu) \leq d_{\max}E_+(\mu)$ and $c_0 > 0$:

$$F(\mu) = \frac{S(\mu)}{E_+(\mu) + c_0} \leq \frac{d_{\max}E_+(\mu)}{E_+(\mu) + c_0} < d_{\max}.$$

**(3) Finite fixed point and monotone convergence for $d_\star < d_{\max}$.** By (2), $F(\cdot)$ is continuous, strictly increasing, with range $(0, d_{\max})$. Therefore, there exists a unique $\mu^\star \in \mathbb{R}$ such that $F(\mu^\star) = d_\star$. Now we first argue $\mu(t) \to \mu^\star$.

Consider the ODE $\dot{\mu} = c\left(d_\star - F(\mu)\right)$ with $c = \alpha/(n\sqrt{K}) > 0$. If $\mu(0) = 0 \leq \mu^\star$ and (thus) $F(\mu(0)) \leq F(\mu^\star) = d_\star$, then $\dot{\mu}(t) \geq 0$ as long as $\mu(t) \leq \mu^\star$, hence $\mu$ is non-decreasing and bounded above by $\mu^\star$; monotone convergence implies $\mu(t) \to \bar{\mu} \leq \mu^\star$ for some $\bar{\mu}$. By the continuity of $F(\cdot)$ and the fact that $\dot{\mu} \to 0$, we can obtain that $F(\bar{\mu}) = d_\star$, which implies $\bar{\mu} = \mu^\star$. The case $\mu(0) > \mu^\star$ is analogous with a non-increasing trajectory.

For local exponential stability, we can set $\tilde{\mu} = \mu - \mu^\star$ and write

$$\dot{\tilde{\mu}}(t) = -c\left(F(\mu^\star + \tilde{\mu}(t)) - F(\mu^\star)\right).$$

By the mean value theorem, $F(\mu^\star + \tilde{\mu}) - F(\mu^\star) = F'(\xi)\,\tilde{\mu}$ for some $\xi$ between $\mu^\star$ and $\mu^\star + \tilde{\mu}$. Since $F'(\mu^\star) > 0$ and $F'$ is continuous, there exists $\eta > 0$ and $m > 0$ such that $F'(\xi) \geq m$ whenever $|\xi - \mu^\star| \leq \eta$. Hence, as long as $|\tilde{\mu}(t)| \leq \eta$, we have

$$\frac{d}{dt}|\tilde{\mu}(t)| = \frac{\tilde{\mu}(t)}{|\tilde{\mu}(t)|}\dot{\tilde{\mu}}(t) = -c\,F'(\xi(t))\,|\tilde{\mu}(t)| \leq -c\,m\,|\tilde{\mu}(t)|.$$

Applying Gronwall's inequality, we have $|\tilde{\mu}(t)| \leq e^{-cmt}|\tilde{\mu}(0)|$ in this neighborhood, which establishes local exponential convergence.

**(4) Divergence and logarithmic lower bound for $d_\star = d_{\max}$.** When $d_\star = d_{\max}$, since $F(\mu) < d_{\max}$ for all finite $\mu$, we have $\dot{\mu}(t) = c\left(d_{\max} - F(\mu(t))\right) > 0$ where $c = \frac{\alpha}{n\sqrt{K}}$ and thus $\mu(t)$ is strictly increasing. We now lower bound the growth rate similar to Lemma 5.

Since $S(\mu) \leq d_{\max}E_+(\mu)$, we have

$$d_{\max} - F(\mu) = d_{\max} - \frac{S(\mu)}{E_+(\mu) + c_0} \geq d_{\max}\left(1 - \frac{E_+(\mu)}{E_+(\mu) + c_0}\right) = \frac{d_{\max}c_0}{E_+(\mu) + c_0}.$$

Moreover, for each $u \in \mathcal{N}_{c+1}$, it holds that

$$e^{\xi_u(\mu)} \leq \exp(\lambda(1 + \mu d_{\max})).$$

Therefore, $E_+(\mu) \leq |\mathcal{N}_{c+1}| e^{\lambda(1+\mu d_{\max})} \leq n\,e^{\lambda(1+\mu d_{\max})}$ and thus we can obtain that

$$E_+(\mu) + c_0 \leq n\,e^{\lambda(1+\mu d_{\max})} + c_0 \leq n\left(e^{\lambda(1+\mu d_{\max})} + 1\right) \leq 2n\,e^{\lambda(1+\mu d_{\max})},$$

where we used $e^x \geq 1$ for $x \geq 0$. Combining the above derivation, we can obtain that

$$d_{\max} - F(\mu) \geq \frac{d_{\max}c_0}{2n}e^{-\lambda}e^{-\lambda d_{\max}\mu}.$$

We can then plug this into $\dot{\mu} = c\left(d_{\max} - F(\mu)\right)$ with $c = \alpha/(n\sqrt{K})$ to get

$$\dot{\mu}(t) \geq \frac{\alpha}{n\sqrt{K}} \cdot \frac{d_{\max}c_0}{2n}e^{-\lambda}e^{-\lambda d_{\max}\mu(t)} = c_1 e^{-c_2\mu(t)},$$

where $c_1 = \frac{\alpha\,d_{\max}c_0\,e^{-\lambda}}{2\,n^2\sqrt{K}}$ and $c_2 = \lambda d_{\max} > 0$.

Applying Lemma 13, we can obtain exactly (13). This shows $\mu(t) \to +\infty$ at least logarithmically fast. □

### B.3 THOUGHT EXPANSION

Finally, we provide results for continuous thought expansion. Note that the following results hold for any directed graph that differs from the graphs in the training set.

**Theorem 5** (One-hop expansion of continuous thoughts). *Let $\mathcal{G} = (\mathcal{V}, \mathcal{E})$ be any directed graph (which can differ from the graphs in the training set) and $r \in \mathcal{V}$ be a root node. Assume the current thought is any positive superposition on $\mathcal{N}_c^{\mathcal{G}}(r)$:*

$$\boldsymbol{h}_{[t_c]} = \sum_{u \in \mathcal{N}_c} \lambda_u \, \mathsf{E}(u), \qquad \lambda_u > 0.$$

*Then the next continuous thought $[t_{c+1}] = \boldsymbol{h}_{[t_{c+1}]}$ generated by the forward pass (2) satisfies*

$$\boldsymbol{\xi} = \mathbf{U}^\top \boldsymbol{h}_{[t_{c+1}]} = \sum_{v \in \mathcal{N}_{c+1}} \beta_v \, \boldsymbol{e}_v,$$

*with coefficients*

$$\beta_v = \underbrace{\lambda_v \, \mathbb{1}\{v \in \mathcal{N}_c\}}_{carryover} + \underbrace{\mu \sum_{u \in \mathcal{N}_c} \lambda_u \, \mathbb{1}\{(u \to v) \in \mathcal{E}\}}_{one\text{-}hop\ expansion} \geq 0. \qquad (14)$$

*where we assume the model has been trained until time $t$ and the trained model satisfies $\mu_v(t) \equiv \mu > 0$ in (3). In particular, $\beta_v > 0$ for every $v \in \mathcal{N}_{c+1}$ if $\mu > 0$, so the support of $\boldsymbol{\xi}$ is exactly $\mathcal{N}_{c+1}$, and the output is a superposition of $\mathcal{N}_{c+1}$.*

*Proof.* Note that $\boldsymbol{h}_i = \mathsf{E}_s(s_i) + \mathsf{E}_t(t_i)$ and $\mathbf{W} = \sum_{v \in \mathcal{V}} \mu_v(t) \mathsf{E}(v) \mathsf{E}_s(v)^\top = \sum_{v \in \mathcal{V}} \mu \mathsf{E}(v) \mathsf{E}_s(v)^\top$. We can calculate that

$$\mathbf{W} \boldsymbol{h}_i = \sum_{v \in \mathcal{V}} \mu \, \mathsf{E}(v) \mathsf{E}_s(v)^\top \big(\mathsf{E}_s(s_i) + \mathsf{E}_t(t_i)\big) = \mu \, \mathsf{E}(s_i),$$

where we used $\mathsf{E}_s(v)^\top \mathsf{E}_s(s_i) = \mathbb{1}\{v = s_i\}$ and $\mathsf{E}_s(v)^\top \mathsf{E}_t(t_i) = 0$ according to Assumption 1. Therefore, with $\boldsymbol{h}_{[t_c]} = \sum_{u \in \mathcal{N}_c} \lambda_u \mathsf{E}(u)$, we can calculate

$$\alpha_i := \boldsymbol{h}_{[t_c]}^\top \mathbf{W} \boldsymbol{h}_i = \mu \sum_{u \in \mathcal{N}_c} \lambda_u \, \mathsf{E}(u)^\top \mathsf{E}(s_i) = \mu \, \lambda_{s_i} \, \mathbb{1}\{s_i \in \mathcal{N}_c\}.$$

The value aggregation becomes

$$\phi(\boldsymbol{h}_{[t_c]}; \{\boldsymbol{h}_i\}) = \sum_{i=1}^m \alpha_i \boldsymbol{h}_i = \mu \sum_{i:\, s_i \in \mathcal{N}_c} \lambda_{s_i} \big(\mathsf{E}_s(s_i) + \mathsf{E}_t(t_i)\big).$$

Furthermore, we have

$$\mathbf{U}^\top \phi(\boldsymbol{h}_{[t_c]}; \{\boldsymbol{h}_i\}) = \mu \sum_{i:\, s_i \in \mathcal{N}_c} \lambda_{s_i} \, \boldsymbol{e}_{t_i} = \mu \sum_{v \in \mathcal{V}} \Big( \sum_{u \in \mathcal{N}_c} \lambda_u \, \mathbb{1}\{(u \to v) \in \mathcal{E}\} \Big) \boldsymbol{e}_v.$$

Similarly,

$$\mathbf{U}^\top \boldsymbol{h}_{[t_c]} = \sum_{v \in \mathcal{N}_c} \lambda_v \, \boldsymbol{e}_u = \sum_{v \in \mathcal{V}} \lambda_v \, \mathbb{1}\{v \in \mathcal{N}_c\} \, \boldsymbol{e}_v.$$

Adding the two parts yields $\boldsymbol{\xi} = \sum_v \beta_v \, \boldsymbol{e}_v$ with $\beta_v$ as in (14), since by definition we have $\boldsymbol{h}_{[t_{c+1}]} = \boldsymbol{h}_{[t_c]} + \phi(\boldsymbol{h}_{[t_c]}; \{\boldsymbol{h}_i\})$ and $\boldsymbol{\xi} = \mathbf{U}^\top \boldsymbol{h}_{[t_{c+1}]}$. □

## C  MISSING PROOFS FOR SECTION 4

In this section, we analyze the training dynamics of the prediction stage, i.e., after thought generation, how the model extracts the information from the generated continuous thought to make the final prediction.

Recall that the $i$-th training sample consists of $(\mathcal{G}^{(i)}, \mathbf{r}^{(i)}, \mathbf{c}_1^{(i)}, \mathbf{c}_2^{(i)}, \boldsymbol{\lambda}^{(i)})$, and we denote $\mathbf{c}_\star^{(i)}$ as the reachable candidate and $\mathbf{c}_\perp^{(i)}$ as the unreachable candidate. We also use $\mathcal{N}_C^{(i)} = \mathcal{N}_C^{\mathcal{G}^{(i)}}(\mathbf{r}^{(i)})$ to denote the $C$-ball for the $i$-th training sample. We assume $C$ is large enough so that $\mathbf{c}_\star^{(i)} \in \mathcal{N}_C^{(i)}$ for any $i$. Note that $\mathbf{c}_\perp^{(i)} \notin \mathcal{N}_C^{(i)}$ for any $C$ by definition. For notation convenience, we also use $\mu_A = \mu_{\texttt{<A>}}$, $\mu_R = \mu_{\texttt{<R>}}$, and denote $\boldsymbol{\xi}^{(i)} = \{\xi_v^{(i)}\}_{v \in \mathsf{Voc}}$ as the logits calculated by forward pass (7) for the $i$-th training sample. We denote $\xi_{\mathbf{c}_t^{(i)}}^{(i)} = \xi_{\mathbf{c}_t}^{(i)}$, $\lambda_{\mathbf{c}_t^{(i)}}^{(i)} = \lambda_{\mathbf{c}_t}^{(i)}$ for $t \in \{1, 2, \star, \perp\}$ for notation convenience.

To start with, we first provide a closed-form logit expression.

**Lemma 8** (Closed-form logits in prediction stage)**.** *Under reparameterization* (8) *and forward pass for the prediction stage* (7)*, for every $v \in \mathcal{V}$ we have*

$$\xi_v(\mu_{\texttt{<A>}}, \mu_{\texttt{<R>}}) = \underbrace{\mu_{\texttt{<A>}} \lambda_v}_{\text{frontier residual}} + \underbrace{\mu_{\texttt{<R>}} \mathbb{1}\{v \in \{c_1, c_2\}\}}_{\text{candidate lift}}. \tag{15}$$

*In particular,*

$$\xi_{c_\star} - \xi_{c_\perp} = \mu_{\texttt{<A>}} \lambda_{c_\star}. \tag{16}$$

*Proof.* For the reasoning token $\boldsymbol{h}_{\mathsf{Idx}(\texttt{<R>})} = \mathsf{E}(\texttt{<R>}) + \mathsf{E}(c_1) + \mathsf{E}(c_2)$, we have

$$\mathbf{W} \boldsymbol{h}_{\mathsf{Idx}(\texttt{<R>})} = \mu_{\texttt{<R>}} \mathsf{E}(\texttt{<A>}) \underbrace{\mathsf{E}(\texttt{<R>})^\top \boldsymbol{h}_{\mathsf{Idx}(\texttt{<R>})}}_{=1} = \mu_{\texttt{<R>}} \mathsf{E}(\texttt{<A>}).$$

Therefore,

$$\mathbf{U}^\top \big( (\boldsymbol{h}_{\mathsf{Idx}(\texttt{<A>})}^\top \mathbf{W} \boldsymbol{h}_{\mathsf{Idx}(\texttt{<R>})}) \boldsymbol{h}_{\mathsf{Idx}(\texttt{<R>})} \big) = \mu_{\texttt{<R>}} (\boldsymbol{e}_{\texttt{<R>}} + \boldsymbol{e}_{c_1} + \boldsymbol{e}_{c_2}).$$

Also,

$$\mathbf{U}^\top \big( \mu_{\texttt{<A>}} \boldsymbol{h}_{\mathsf{Idx}(\texttt{<A>})} \big) = \mu_{\texttt{<A>}} \sum_{u \in \mathcal{N}_C} \lambda_u \boldsymbol{e}_u + \mu_{\texttt{<A>}} \boldsymbol{e}_{\texttt{<A>}}.$$

Combining the above two expressions yields (15). For $c_\perp \notin \mathcal{N}_C$ we have $\lambda_{c_\perp} = 0$, and (16) follows. $\square$

For each training sample, we can construct a two-dimensional feature for every node $v \in \mathcal{V}$:

$$x_v^{(i)} := \big( \lambda_v^{(i)}, \ \mathbb{1}\{v \in \{c_1^{(i)}, c_2^{(i)}\}\} \big) \in \mathbb{R}_{\geq 0}^2,$$

so that $\xi_v^{(i)}(\mu_A, \mu_R) = \langle w, x_v^{(i)} \rangle$ with $w := (\mu_A, \mu_R) \in \mathbb{R}_{\geq 0}^2$. For instance $i$, a correct classification means $\langle w, x_{c_\star}^{(i)} - x_v^{(i)} \rangle > 0$ for all $v \neq c_\star^{(i)}$, where we denote $x_{c_t^{(i)}}^{(i)} = x_{c_t}^{(i)}$ for $t \in \{1, 2, \star, \perp\}$. We further define the difference of features with respect to $c_\star$ for later use:

$$\Delta_{i,v} := x_{c_\star}^{(i)} - x_v^{(i)} = \begin{cases} (\lambda_{c_\star}^{(i)}, 0), & v = c_\perp^{(i)}, \\ (\lambda_{c_\star}^{(i)} - \lambda_v^{(i)}, 1), & v \in \mathcal{N}_C^{(i)} \setminus \{c_\star^{(i)}\}, \\ (\lambda_{c_\star}^{(i)}, 1), & v \notin \mathcal{N}_C^{(i)} \cup \{c_\perp^{(i)}\}. \end{cases} \tag{17}$$

## C.1 LINEARLY SEPARABLE STRUCTURE AND A MAX-MARGIN PROBLEM

**Lemma 9** (Separation by a nonnegative direction)**.** *For every instance $i$ and each competitor $v \neq c_\star^{(i)}$,*

$$\langle (1, 1), \Delta_{i,v} \rangle = \begin{cases} \lambda_{c_\star}^{(i)}, & v = c_\perp^{(i)}, \\ \lambda_{c_\star}^{(i)} - \lambda_v^{(i)} + 1, & v \in \mathcal{N}_C^{(i)} \setminus \{c_\star^{(i)}\}, \\ \lambda_{c_\star}^{(i)} + 1, & v \notin \mathcal{N}_C^{(i)} \cup \{c_\perp^{(i)}\}, \end{cases} > 0.$$

*Hence, the training data are linearly separable by a direction in $\mathbb{R}_{\geq 0}^2$.*

*Proof.* The result holds because $\lambda_{c_\star}^{(i)} > 0$, $\lambda_v^{(i)} \leq 1$. $\square$

Define the *hard-margin* value of a unit direction $u \in \mathbb{S}^1 \cap \mathbb{R}^2_{\geq 0}$ (where $\mathbb{S}^1 = \{u \in \mathbb{R}^2 : \|u\|_2 = 1\}$) as

$$\gamma(u) := \min_i \min_{v \neq c^{(i)}_\star} \langle u, \Delta_{i,v} \rangle.$$

The corresponding *maximum-margin* direction is

$$u^\star \in \arg\max_{u \in \mathbb{S}^1 \cap \mathbb{R}^2_{\geq 0}} \gamma(u). \tag{18}$$

We characterize $u^\star$ using the following two quantities of the training sets:

$$\lambda_\star := \min_i \lambda^{(i)}_{c_\star} \in (0, 1], \qquad \Delta_{\text{train}} := \max_i \max_{v \in \mathcal{N}^{(i)}_C \setminus \{c^{(i)}_\star\}} \left(\lambda^{(i)}_v - \lambda^{(i)}_{c_\star}\right)_+ \in [0, 1],$$

where $(x)_+ := \max\{x, 0\}$. Intuitively, $\lambda_\star$ is the smallest mass ever placed on a reachable candidate across the training set, and $\Delta_{\text{train}}$ is the largest overshoot of a non-candidate but reachable node's weight relative to the reachable candidate.

**Lemma 10** (Closed-form lower envelope of the margin). *For any unit $u = (u_A, u_R) \in \mathbb{S}^1 \cap \mathbb{R}^2_{\geq 0}$,*

$$\gamma(u) = \min\{u_A \lambda_\star, \ u_R - u_A \Delta_{\text{train}}, \ u_A \lambda_\star + u_R\} = \min\{u_A \lambda_\star, \ u_R - u_A \Delta_{\text{train}}\}.$$

*Proof.* According to (17), we can directly obtain the the lower bounds $u_A \lambda^{(i)}_{c_\star}$, $u_R + u_A(\lambda^{(i)}_{c_\star} - \lambda^{(i)}_v)$, and $u_A \lambda^{(i)}_{c_\star} + u_R$. Minimizing over $i$ and $v$ according to the definition of $\lambda_\star, \Delta_{\text{train}}$ yields the desired result. $\square$

**Proposition C.1** (Properties of the maximum-margin direction). *Let $u^\star = (u^\star_A, u^\star_R)$ be a solution of* (18). *Then the unique maximizer satisfies*

$$\frac{u^\star_R}{u^\star_A} = \lambda_\star + \Delta_{\text{train}}, \qquad u^\star_A = \frac{1}{\sqrt{1 + (\lambda_\star + \Delta_{\text{train}})^2}}, \qquad u^\star_R = \frac{\lambda_\star + \Delta_{\text{train}}}{\sqrt{1 + (\lambda_\star + \Delta_{\text{train}})^2}}.$$

*Proof.* By Lemma 10, we can maximize $\gamma(u) = \min\{u_A \lambda_\star, \ u_R - u_A \Delta_{\text{train}}\}$ over the unit vector $u$ by equalizing the two arguments (otherwise one can rotate $u$ to increase the minimum). Therefore, we can equalize the two arguments, which yields $u_R = u_A(\lambda_\star + \Delta_{\text{train}})$, and obtain the desired result. $\square$

## C.2 IMPLICIT BIAS OF GRADIENT FLOW AND DIRECTIONAL CONVERGENCE

$$\ell^{\text{pred}}_{\mathcal{G}, r, c_1, c_2, \boldsymbol{\lambda}} := -\log \frac{\exp(\xi_{c_\star})}{\sum_{v \in \mathcal{V}} \exp(\xi_v)}, \qquad \mathcal{L}^{\text{pred}} = \mathbb{E}_{(\mathcal{G}, r, c_1, c_2, \boldsymbol{\lambda}) \sim \mathcal{D}}[\ell^{\text{pred}}_{\mathcal{G}, r, c_1, c_2, \boldsymbol{\lambda}}],$$

Recall the loss function on the prediction stage over the training set (9). We can rewrite it as follows

$$\mathcal{L}(\mu_A, \mu_R) := \frac{1}{N} \sum_{i=1}^N \ell^{(i)}(\mu_A, \mu_R), \qquad \ell^{(i)}(\mu_A, \mu_R) := -\log \frac{\exp(\xi^{(i)}_{c_\star^{(i)}})}{\sum_{v \in \mathcal{V}} \exp(\xi^{(i)}_v)},$$

and run the gradient-flow dynamics $\dot{w}(t) = -\alpha \nabla \mathcal{L}(w(t))$ with $w(t) = (\mu_A(t), \mu_R(t))$ and $\alpha > 0$. By Lemma 9, the data are linearly separable, so the implicit bias of gradient flow directly yields the following lemma.

**Lemma 11** (Implicit bias of gradient flow). *Along gradient flow from any bounded initialization $w(0)$, we have*

$$\|w(t)\| \to \infty, \qquad \frac{w(t)}{\|w(t)\|} \to u^\star,$$

*where $u^\star$ is the unique solution to the maximum-margin problem* (18). *Combining Proposition C.1, there exists a scalar radius $r(t) \to \infty$ such that*

$$(\mu_A(t), \mu_R(t)) = r(t) u^\star + o(r(t)),$$

*and for any $\varepsilon > 0$,*

$$\frac{\mu_R(t)}{\mu_A(t)} \geq \lambda_\star + \Delta_{\text{train}} - \varepsilon \quad \text{for all sufficiently large } t.$$

The proof can be straightforwardly adapted from its gradient descent counterpart (Soudry et al., 2018).

## C.3 PREDICTION ON UNSEEN GRAPHS

Finally, we show that after sufficient training, the model can correctly predict the reachable candidate node even for unseen graphs, showcasing its generalization capability.

Fix any unseen test graph along with the exploration set $\mathcal{N}_C^{\text{test}}$ and weights $\boldsymbol{\lambda}^{\text{test}}$, such that $\lambda_v^{\text{test}} \in (0, 1]$ on $\mathcal{N}_C^{\text{test}}$ and $0$ otherwise. The test graph also satisfies

$$\max_{u \in \mathcal{N}_C^{\text{test}}} \lambda_u^{\text{test}} - \lambda_{\mathrm{c}_\star}^{\text{test}} \leq \Delta, \quad \text{with} \quad \Delta \leq \Delta_{\text{train}}.$$

Therefore, for every non-candidate $v \in \mathcal{N}_C^{\text{test}} \setminus \{\mathrm{c}_\star^{\text{test}}\}$, it holds that $\lambda_v^{\text{test}} \leq \lambda_{\mathrm{c}_\star}^{\text{test}} + \Delta$.

The following lemma shows that as long as the test graph satisfies the above condition, it has a positive margin using the maximum margin direction for the training set $u^\star$.

**Lemma 12** (Positive test-time margins from the trained direction). *Let $u^\star = (u_A^\star, u_R^\star)$ be the unique max-margin direction with $u_R^\star / u_A^\star = \lambda_\star + \Delta_{\text{train}} > \Delta$. Then for every competitor $v \neq \mathrm{c}_\star^{\text{test}}$,*

$$\langle u^\star, \, x_{\mathrm{c}_\star}^{\text{test}} - x_v^{\text{test}} \rangle \geq \min\{u_A^\star \lambda_\star, \ u_A^\star \lambda_\star^{\text{test}}\} > 0.$$

*Proof.* For $v = \mathrm{c}_\perp^{\text{test}}$, the difference is $(\lambda_{\mathrm{c}_\star}^{\text{test}}, 0)$; since $\lambda_{\mathrm{c}_\star}^{\text{test}} > 0$ we have $\langle u^\star, x_{\mathrm{c}_\star}^{\text{test}} - x_{\mathrm{c}_\perp}^{\text{test}} \rangle \geq u_A^\star \lambda_{\mathrm{c}_\star}^{\text{test}} > 0$.

For $v \notin \mathcal{N}_C^{\text{test}}$ the difference is $(\lambda_{\mathrm{c}_\star}^{\text{test}}, 1)$ and the bound is even larger.

For $v \in \mathcal{N}_C^{\text{test}} \setminus \{\mathrm{c}_\star^{\text{test}}\}$, we have $\lambda_v^{\text{test}} \leq \lambda_{\mathrm{c}_\star}^{\text{test}} + \Delta$, hence

$$\langle u^\star, \, x_{\mathrm{c}_\star}^{\text{test}} - x_v^{\text{test}} \rangle = u_A^\star \big( \lambda_{\mathrm{c}_\star}^{\text{test}} - \lambda_v^{\text{test}} \big) + u_R^\star \geq u_R^\star - u_A^\star \Delta \geq u_A^\star \big( \lambda_\star + \Delta_{\text{train}} - \Delta \big) \geq u_A^\star \lambda_\star > 0.$$

$\square$

Finally, we show that after sufficient training, the model can correctly predict the reachable candidate node .

**Theorem 6** (Generalization for unseen graphs). *Let $(\mu_A(t), \mu_R(t))$ follow gradient flow on the loss (9) from any bounded initialization. Suppose the training set is linearly separable and $\lambda_\star, \Delta_{\text{train}}$ are defined as above. Then, for any unseen instance satisfying $\lambda_v^{\text{test}} \in (0, 1]$ on $\mathcal{N}_C^{\text{test}}$ and $0$ otherwise, and*

$$\max_{u \in \mathcal{N}_C^{\text{test}}} \lambda_u^{\text{test}} - \lambda_{\mathrm{c}_\star}^{\text{test}} \leq \Delta, \quad \text{with} \quad \Delta \leq \Delta_{\text{train}},$$

*we have for all sufficiently large $t$:*

$$p_{\mathrm{c}_\star^{\text{test}}}(t) := \frac{\exp\big(\xi_{\mathrm{c}_\star^{\text{test}}}^{\text{test}}(\mu_A(t), \mu_R(t))\big)}{\sum_v \exp\big(\xi_v^{\text{test}}(\mu_A(t), \mu_R(t))\big)} \to 1.$$

*Proof.* By Lemma 11, we have

$$(\mu_A(t), \mu_R(t)) = r(t) u^\star + o(r(t)), \quad r(t) \to \infty.$$

Then, by Lemma 12, for every competitor $v \neq \mathrm{c}_\star^{\text{test}}$,

$$\begin{aligned} & \xi_{\mathrm{c}_\star}^{\text{test}}(\mu_A(t), \mu_R(t)) - \xi_v^{\text{test}}(\mu_A(t), \mu_R(t)) \\ =& r(t) \langle u^\star, \, x_{\mathrm{c}_\star}^{\text{test}} - x_v^{\text{test}} \rangle + o\big(r(t)\big) \\ \geq& r(t) \cdot \min\{u_A^\star \lambda_\star, \ u_A^\star \lambda_\star^{\text{test}}\} + o\big(r(t)\big) \xrightarrow[t \to \infty]{} +\infty. \end{aligned}$$

Hence the $\arg\max$ is $\mathrm{c}_\star^{\text{test}}$, and its softmax probability tends to 1. $\square$

# D AUXILIARY LEMMAS

**Lemma 13** (ODE lower bound). *Let $c_1, c_2 > 0$ be two constants. Assume the function $f : \mathbb{R} \to \mathbb{R}$ satisfies $f(0) = 0$ and*

$$\frac{df(t)}{dt} \geq c_1 \cdot \exp(-c_2 \cdot f(t)), \quad \forall t \geq 0.$$

*Then it holds*

$$f(t) \geq \frac{1}{c_2} \ln(1 + c_1 c_2 t)$$

*for all $t \geq 0$.*

*Proof.* We define $g(t) = e^{c_2 f(t)}$. Note that

$$\frac{dg(t)}{dt} = \frac{d}{dt}(e^{c_2 f(t)}) = c_2 \frac{df(t)}{dt} \cdot e^{c_2 f(t)} \geq c_2 \cdot c_1 \cdot \exp(-c_2 \cdot f(t)) \exp(c_2 f(t)) = c_1 c_2.$$

Therefore, $dg(t) \geq c_1 c_2 dt$ for $t \geq 0$, and thus

$$\int_0^t dg(t) \geq \int_0^t c_1 c_2 dt \implies g(t) - g(0) \geq c_1 c_2 t.$$

Therefore,

$$g(t) = e^{c_2 f(t)} \geq g(0) + c_1 c_2 t = e^{c_2 f(0)} + c_1 c_2 t = 1 + c_1 c_2 t,$$

which implies

$$f(t) \geq \frac{1}{c_2} \ln(1 + c_1 c_2 t).$$

$\square$

# E EXPERIMENT DETAILS

## E.1 DATASET

Table 3: ProsQA statistics. Numbers are averaged over problem instances.

|       | #Problems | $|V|$ | $|E|$ | Sol. Len. |
|-------|-----------|-------|-------|-----------|
| Train | 14785     | 22.8  | 36.5  | 3.5       |
| Val   | 257       | 22.7  | 36.3  | 3.5       |
| Test  | 419       | 22.7  | 36.0  | 3.5       |

The statistics of the ProsQA dataset is shown in Table 3.

## E.2 EXPERIMENT WITH COCONUT-BFS

As a comparison, we also train a model with a modified version of $\mathcal{L}^{\text{BFS}}$. Recall that the original $\mathcal{L}^{\text{BFS}}$ (5) encourages the model to predict any nodes within $\mathcal{N}_{c+1}$. To avoid the trivial solution of always predicting the root node, we introduce an experimental variant that only encourages predicting nodes on the current frontier:

$$\textbf{COCONUT-BFS-exp:} \quad \ell_{\mathcal{G}, \text{r}}^{\text{BFS-exp}} := -\log \frac{\sum_{v \in \mathcal{N}_{c+1} \backslash \mathcal{N}_c} \exp(\xi_v)}{\sum_{v \in \mathcal{V}} \exp(\xi_v)}. \tag{19}$$

All other training settings remain unchanged. The answer accuracy of this model on the test set is 99.0%. We then track the logit difference between frontier and non-frontier edges as a proxy for $\mu_v$, with results shown in Figure 5.

| Number of layers | |
|---|---|
| $L = 2$ | 98.8 |
| $L = 4$ | 97.3 |
| $L = 8$ | 96.5 |
| $L = 12$ | 67.4 |

| Number of heads | |
|---|---|
| $H = 4$ | 98.0 |
| $H = 8$ | 98.8 |
| $H = 12$ | 98.8 |

| Width | |
|---|---|
| $d_{\text{model}} = 384$ | 62.0 |
| $d_{\text{model}} = 768$ | 98.8 |
| $d_{\text{model}} = 1536$ | 97.7 |

| Learning rate | |
|---|---|
| $\eta = 2 \times 10^{-4}$ | 58.1 |
| $\eta = 1 \times 10^{-4}$ | 98.8 |
| $\eta = 5 \times 10^{-5}$ | 62.1 |

| Weight tying | |
|---|---|
| Tied | 98.8 |
| Untied | 98.8 |

Table 4: Ablation on depth, heads, width, learning rate, and weight tying. By default, other hyper-parameters follow the main experiments.

In Stage 1, the logit difference for $c = 1$ grows much faster than under $\mathcal{L}^{\text{coco}}$ and shows no sign of saturation even after 150 epochs. This agrees with the theoretical prediction in Theorem 1: under COCONUT-BFS, $\mu_v$ diverges rather than stabilizing. At later steps ($c = 3, 4$), the gap between COCONUT-BFS and COCONUT becomes smaller. We attribute this to practical factors such as stage-wise data mixing, gradient propagation across earlier thoughts, and a larger discrepancy between losses (5) and (19) in the later stage.

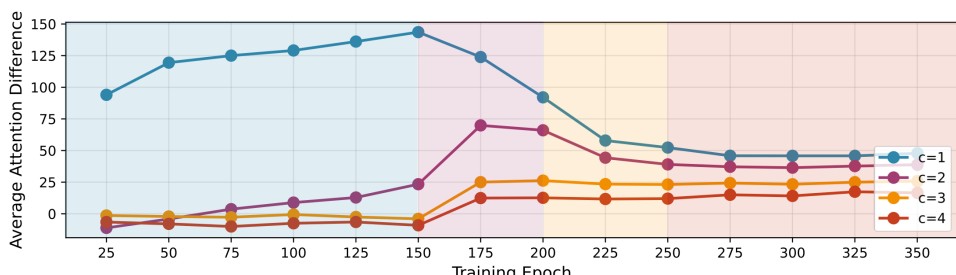

Figure 5: The attention logits difference between frontier edges and others. The model is trained with a modified version of $\mathcal{L}^{\text{BFS}}$.

### E.3 ALTERNATIVE ATTENTION ROUTES FOR CANDIDATE LIFT

Our theoretical analysis in Lemma 2 assumes that `<R>` copies the candidate nodes in the first layer, and `<A>` then attends to `<R>` in the second layer. In practice, however, we observe three distinct yet functionally equivalent attention routes that realize the same *candidate lift*. Example attention maps for each route are shown in Figure 6.

## F ADDITIONAL EXPERIMENTAL RESULTS

In this section, we provide additional experiments to complement our main analysis of training dynamics.

### F.1 ABLATION STUDY: ARCHITECTURAL AND OPTIMIZATION SENSITIVITY

We evaluate the sensitivity of COCONUT training to model depth, number of attention heads, hidden width, and learning rate. The results are summarized in Table 4.

We observe that models with $L = \{4, 8\}$ layers maintain high accuracy, while $L = 12$ is harder to optimize. The performance remains comparable when $d_{\text{model}} \in \{768, 1536\}$, but degrades when the width is too small (e.g., $d_{\text{model}} = 384$). Varying the number of heads does not have major effects on

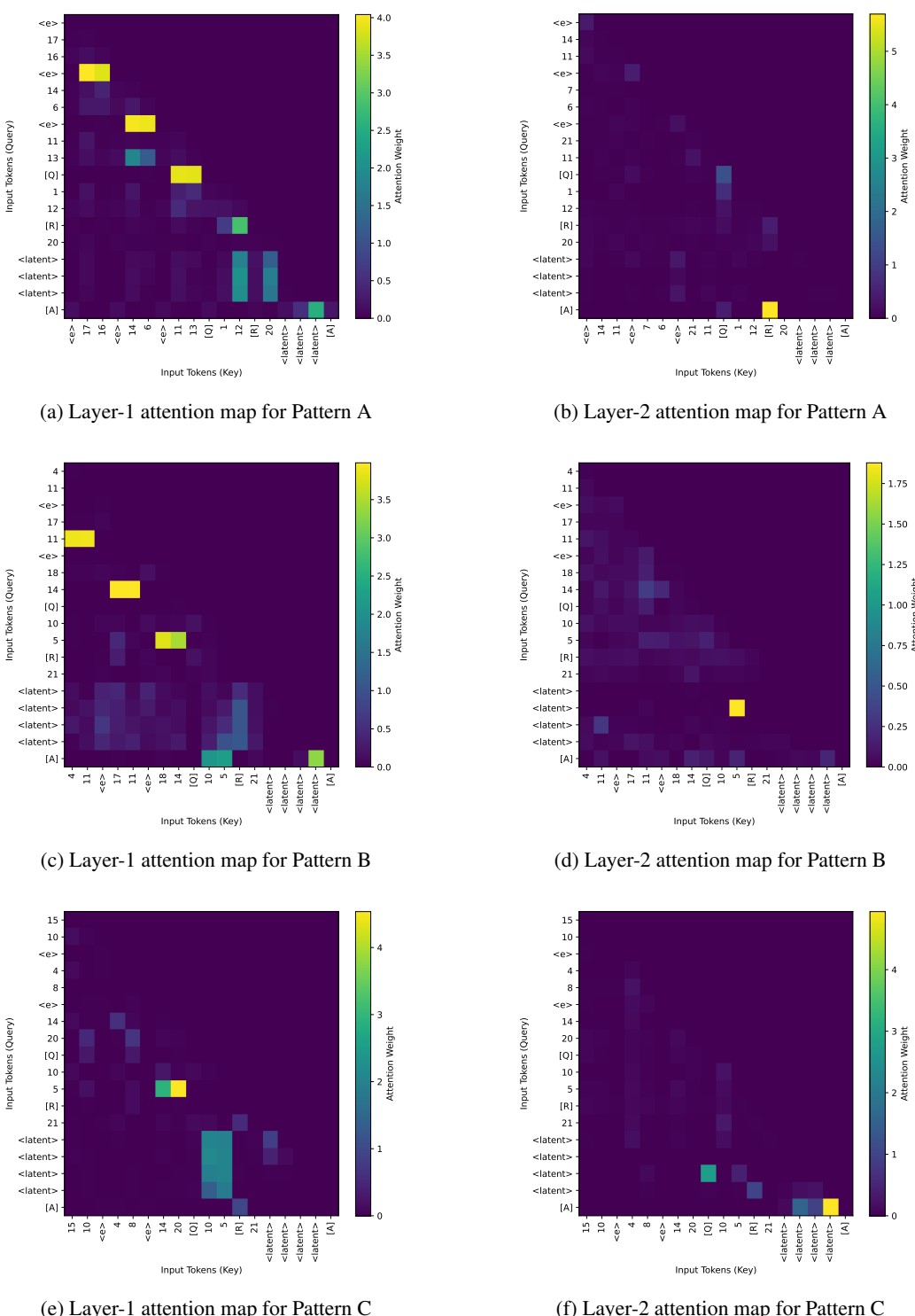

(a) Layer-1 attention map for Pattern A

(b) Layer-2 attention map for Pattern A

(c) Layer-1 attention map for Pattern B

(d) Layer-2 attention map for Pattern B

(e) Layer-1 attention map for Pattern C

(f) Layer-2 attention map for Pattern C

Figure 6: Example attention maps illustrating three alternative routes for *candidate lift*. For clarity, we omit earlier tokens in the sequence and only visualize the final segment containing some of edges, the candidate nodes, latent thoughts, and answer tokens. **Pattern A** (consistent with the theoretical assumption): <R> copies candidate nodes in Layer 1, and <A> attends to <R> in Layer 2. **Pattern B**: <A> directly attends to candidate nodes in Layer 1. **Pattern C**: continuous thoughts copy candidate nodes in Layer 1, and <A> attends to the continuous thoughts in Layer 2. All three patterns achieve the same functional effect of lifting the reachable candidate.

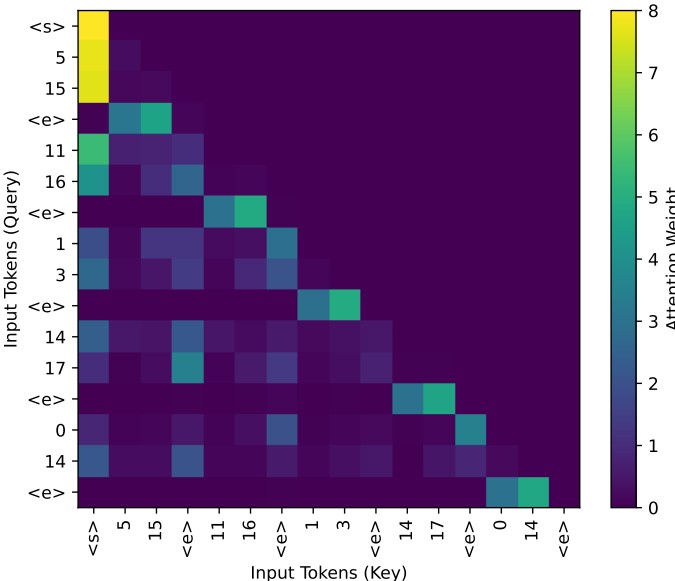

Figure 7: The first-layer attention patterns in 4-layer transformers. `<e>` tokens attend to the corresponding source and target nodes to aggregate the information. This is consistent with the analysis of the two-layer transformer in Zhu et al. (2025).

final accuracy, whereas too large or too small learning rates tend to degrade performance. Weight tying setting does not affect model performance.

We emphasize that each ablation in Table 4 varies only a single hyperparameter at a time, keeping all other settings identical to our main experiment. In practice, these hyperparameters interact in a coupled manner. For instance, with a smaller learning rate of $5 \times 10^{-5}$, we can extend the first-stage training to 300 epochs and get 97.0% accuracy. For deeper models with $L = 12$, prolonging first-stage training to 400 epochs and reducing the learning rate to $5 \times 10^{-5}$ improves accuracy to 99.6%. A comprehensive hyperparameter interaction study is beyond the scope of this work and is left for future investigation.

### F.2 MULTI-LAYER TRANSFORMERS AND MECHANISTIC PATTERNS

We use the COCONUT model with $L = 4$ to analyze the reasoning pattern beyond two-layer transformers. The results are shown in Figure 7 and Figure 8, and we summarize the reasoning patterns below.

- **First layer (induction head):** The first layer performs token-level copying, propagating node information into edge tokens `<e>`, consistent with the copy mechanism derived in previous theoretical analysis (Zhu et al., 2025).

- **Second layer and beyond (superposition):** From the second layer onward, the model aggregates over reachable nodes in a superpositional representation that enables parallel breadth-first exploration.

### F.3 ACCURACY DYNAMICS IN THE ANSWER-PREDICTION STAGE

We track the test accuracy during the final answer-prediction stage following the setting in Figure 4. The result is shown in Figure 9, which shows a rapid transition from near-random guessing to stable high accuracy once the model integrates residual carryover and candidate lift signals.

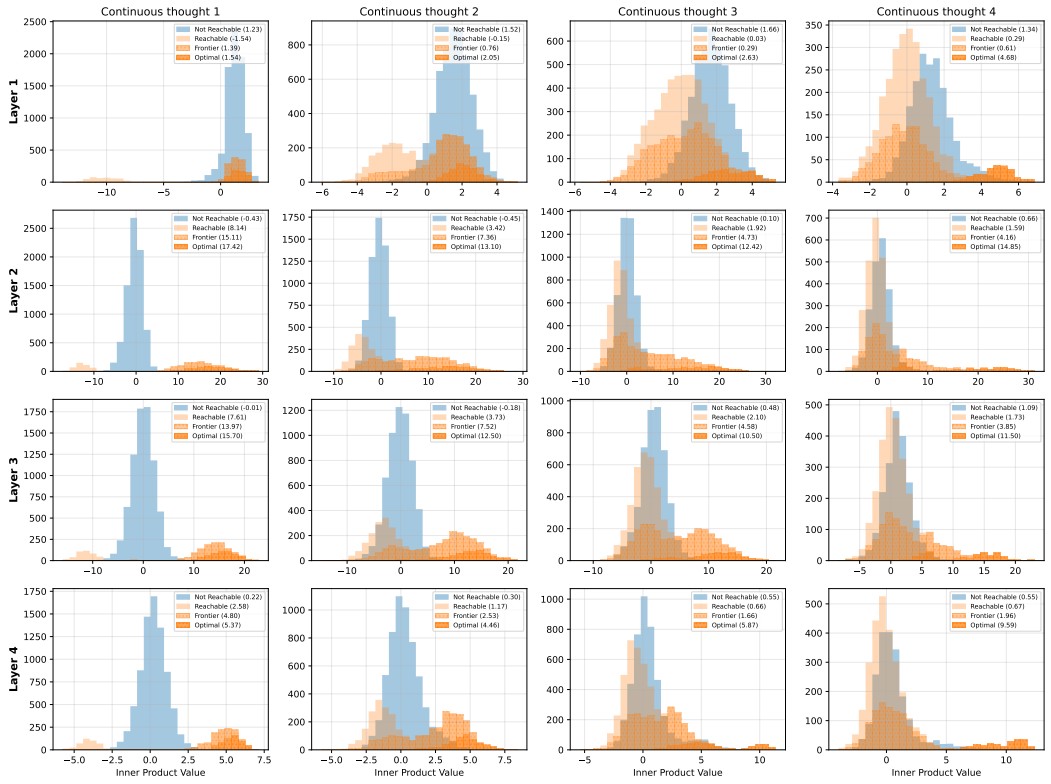

Figure 8: Inner product between layer-wise hidden states and different types of nodes in a 4-layer transformer. The experimental setting follows Zhu et al. (2025). From the second layer onward, hidden states exhibit larger inner products with reachable, frontier, and optimal nodes, indicating that superpositional representations emerge as early as layer 2 in the 4-layer transformer.

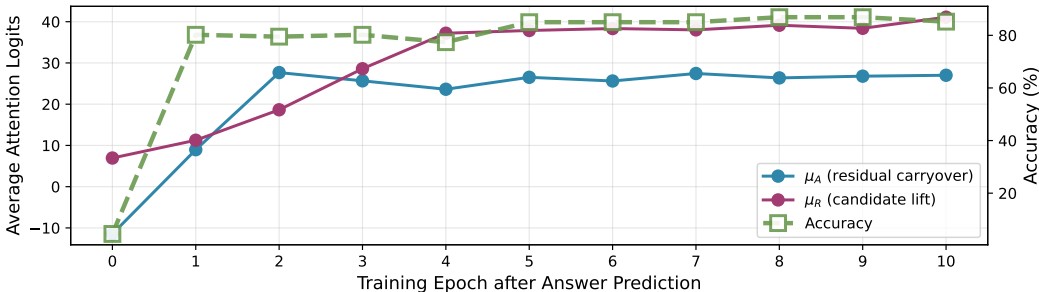

Figure 9: Accuracy curve during the answer-prediction stage. The accuracy shows a rapid improvement corresponding to the learning of residual carryover and candidate lift signals.

# G  THE USE OF LARGE LANGUAGE MODELS (LLMS)

We used LLMs mainly for grammar checking and polishing in paper writing.

