}_\mathrm{s}(v)^\top \big( \mathsf{E}_\mathrm{s}(\mathrm{s}_i) + \mathsf{E}_\mathrm{t}(\mathrm{t}_i) \big) = \mu \, \mathsf{E}(\mathrm{s}_i),$$

where we used $\mathsf{E}_\mathrm{s}(v)^\top \mathsf{E}_\mathrm{s}(\mathrm{s}_i) = \mathbb{1}\{v = \mathrm{s}_i\}$ and $\mathsf{E}_\mathrm{s}(v)^\top \mathsf{E}_\mathrm{t}(\mathrm{t}_i) = 0$ according to Assumption 1. Therefore, with $\boldsymbol{h}_{[t_c]} = \sum_{u \in \mathcal{N}_c} \lambda_u \mathsf{E}(u)$, we can calculate

$$\alpha_i := \boldsymbol{h}_{[t_c]}^\top \mathbf{W} \boldsymbol{h}_i = \mu \sum_{u \in \mathcal{N}_c} \lambda_u \, \mathsf{E}(u)^\top \mathsf{E}(\mathrm{s}_i) = \mu \, \lambda_{\mathrm{s}_i} \, \mathbb{1}\{\mathrm{s}_i \in \mathcal{N}_c\}.$$

The value aggregation becomes

$$\phi(\boldsymbol{h}_{[t_c]}; \{\boldsymbol{h}_i\}) = \sum_{i=1}^{m} \alpha_i \boldsymbol{h}_i = \mu \sum_{i:\, \mathrm{s}_i \in \mathcal{N}_c} \lambda_{\mathrm{s}_i} \big( \mathsf{E}_\mathrm{s}(\mathrm{s}_i) + \mathsf{E}_\mathrm{t}(\mathrm{t}_i) \big).$$

Furthermore, we have

$$\mathbf{U}^\top \phi(\boldsymbol{h}_{[t_c]}; \{\boldsymbol{h}_i\}) = \mu \sum_{i:\, \mathrm{s}_i \in \mathcal{N}_c} \lambda_{\mathrm{s}_i} \, \boldsymbol{e}_{\mathrm{t}_i} = \mu \sum_{v \in \mathcal{V}} \Big( \sum_{u \in \mathcal{N}_c} \lambda_u \, \mathbb{1}\{(u \to v) \in \mathcal{E}\} \Big) \boldsymbol{e}_v.$$

Similarly,

$$\mathbf{U}^\top \boldsymbol{h}_{[t_c]} = \sum_{v \in \mathcal{N}_c} \lambda_v \, \boldsymbol{e}_u = \sum_{v \in \mathcal{V}} \lambda_v \, \mathbb{1}\{v \in \mathcal{N}_c\} \, \boldsymbol{e}_v.$$

Adding the two parts yields $\boldsymbol{\xi} = \sum_v \beta_v \, \boldsymbol{e}_v$ with $\beta_v$ as in (14), since by definition we have $\boldsymbol{h}_{[t_{c+1}]} = \boldsymbol{h}_{[t_c]} + \phi(\boldsymbol{h}_{[t_c]}; \{\boldsymbol{h}_i\})$ and $\boldsymbol{\xi} = \mathbf{U}^\top \boldsymbol{h}_{[t_{c+1}]}$. $\qquad \square$

## C MISSING PROOFS FOR SECTION 4

In this section, we analyze the training dynamics of the prediction stage, i.e., after thought generation, how the model extracts the information from the generated continuous thought to make the final prediction.

Recall that the $i$-th training sample consists of $(\mathcal{G}^{(i)}, \mathtt{r}^{(i)}, \mathtt{c}_1^{(i)}, \mathtt{c}_2^{(i)}, \boldsymbol{\lambda}^{(i)})$, and we denote $\mathtt{c}_\star^{(i)}$ as the reachable candidate and $\mathtt{c}_\perp^{(i)}$ as the unreachable candidate. We also use $\mathcal{N}_C^{(i)} = \mathcal{N}_C^{\mathcal{G}^{(i)}}(\mathtt{r}^{(i)})$ to denote the $C$-ball for the $i$-th training sample. We assume $C$ is large enough so that $\mathtt{c}_\star^{(i)} \in \mathcal{N}_C^{(i)}$ for any $i$. Note that $\mathtt{c}_\perp^{(i)} \notin \mathcal{N}_C^{(i)}$ for any $C$ by definition. For notation convenience, we also use $\mu_A = \mu_{\texttt{<A>}}$, $\mu_R = \mu_{\texttt{<R>}}$, and denote $\boldsymbol{\xi}^{(i)} = \{\xi_v^{(i)}\}_{v \in \mathsf{Voc}}$ as the logits calculated by forward pass (7) for the $i$-th training sample. We denote $\xi_{\mathtt{c}_t^{(i)}}^{(i)} = \xi_{\mathtt{c}_t}^{(i)}$, $\lambda_{\mathtt{c}_t^{(i)}}^{(i)} = \lambda_{\mathtt{c}_t}^{(i)}$ for $t \in \{1, 2, \star, \perp\}$ for notation convenience.

To start with, we first provide a closed-form logit expression.

**Lemma 8** (Closed-form logits in prediction stage). *Under reparameterization* (8) *and forward pass for the prediction stage* (7)*, for every* $v \in \mathcal{V}$ *we have*

$$\xi_v(\mu_{\texttt{<A>}}, \mu_{\texttt{<R>}}) \;=\; \underbrace{\mu_{\texttt{<A>}} \lambda_v}_{\text{frontier residual}} \;+\; \underbrace{\mu_{\texttt{<R>}} \mathbb{1}\{v \in \{c_1, c_2\}\}}_{\text{candidate lift}}. \tag{15}$$

*In particular,*

$$\xi_{c_\star} - \xi_{c_\perp} \;=\; \mu_{\texttt{<A>}} \lambda_{c_\star}. \tag{16}$$

*Proof.* For the reasoning token $\boldsymbol{h}_{\mathsf{Idx}(\texttt{<R>})} = \mathsf{E}(\texttt{<R>}) + \mathsf{E}(\mathtt{c}_1) + \mathsf{E}(\mathtt{c}_2)$, we have

$$\mathbf{W} \boldsymbol{h}_{\mathsf{Idx}(\texttt{<R>})} = \mu_{\texttt{<R>}} \mathsf{E}(\texttt{<A>}) \underbrace{\mathsf{E}(\texttt{<R>})^\top \boldsymbol{h}_{\mathsf{Idx}(\texttt{<R>})}}_{=1} = \mu_{\texttt{<R>}} \mathsf{E}(\texttt{<A>}).$$

Therefore,

$$\mathbf{U}^\top \big( (\boldsymbol{h}_{\mathsf{Idx}(\texttt{<A>})}^\top \mathbf{W} \boldsymbol{h}_{\mathsf{Idx}(\texttt{<R>})}) \boldsymbol{h}_{\mathsf{Idx}(\texttt{<R>})} \big) = \mu_{\texttt{<R>}} (\boldsymbol{e}_{\texttt{<R>}} + \boldsymbol{e}_{\mathtt{c}_1} + \boldsymbol{e}_{\mathtt{c}_2}).$$

Also,

$$\mathbf{U}^\top \big( \mu_{\texttt{<A>}} \boldsymbol{h}_{\mathsf{Idx}(\texttt{<A>})} \big) = \mu_{\texttt{<A>}} \sum_{u \in \mathcal{N}_C} \lambda_u \boldsymbol{e}_u + \mu_{\texttt{<A>}} \boldsymbol{e}_{\texttt{<A>}}.$$

Combining the above two expressions yields (15). For $\mathtt{c}_\perp \notin \mathcal{N}_C$ we have $\lambda_{\mathtt{c}_\perp} = 0$, and (16) follows. $\qquad \square$

For each training sample, we can construct a two-dimensional feature for every node $v \in \mathcal{V}$:

$$x_v^{(i)} \;:=\; \big( \lambda_v^{(i)}, \; \mathbb{1}\{v \in \{\mathtt{c}_1^{(i)}, \mathtt{c}_2^{(i)}\}\} \big) \in \mathbb{R}_{\geq 0}^2,$$

so that $\xi_v^{(i)}(\mu_A, \mu_R) = \langle w, x_v^{(i)} \rangle$ with $w := (\mu_A, \mu_R) \in \mathbb{R}_{\geq 0}^2$. For instance $i$, a correct classification means $\langle w, x_{\mathtt{c}_\star}^{(i)} - x_v^{(i)} \rangle > 0$ for all $v \neq \mathtt{c}_\star^{(i)}$, where we denote $x_{\mathtt{c}_t^{(i)}}^{(i)} = x_{\mathtt{c}_t}^{(i)}$ for $t \in \{1, 2, \star, \perp\}$. We further define the difference of features with respect to $\mathtt{c}_\star$ for later use:

$$\Delta_{i,v} \;:=\; x_{\mathtt{c}_\star}^{(i)} - x_v^{(i)} = \begin{cases} (\lambda_{\mathtt{c}_\star}^{(i)}, 0), & v = \mathtt{c}_\perp^{(i)}, \\ (\lambda_{\mathtt{c}_\star}^{(i)} - \lambda_v^{(i)}, 1), & v \in \mathcal{N}_C^{(i)} \setminus \{\mathtt{c}_\star^{(i)}\}, \\ (\lambda_{\mathtt{c}_\star}^{(i)}, 1), & v \notin \mathcal{N}_C^{(i)} \cup \{\mathtt{c}_\perp^{(i)}\}. \end{cases} \tag{17}$$

### C.1   Linearly separable structure and a max-margin problem

**Lemma 9** (Separation by a nonnegative direction). *For every instance* $i$ *and each competitor* $v \neq \mathtt{c}_\star^{(i)}$,

$$\langle (1,1), \Delta_{i,v} \rangle = \begin{cases} \lambda_{\mathtt{c}_\star}^{(i)}, & v = \mathtt{c}_\perp^{(i)}, \\ \lambda_{\mathtt{c}_\star}^{(i)} - \lambda_v^{(i)} + 1, & v \in \mathcal{N}_C^{(i)} \setminus \{\mathtt{c}_\star^{(i)}\}, \\ \lambda_{\mathtt{c}_\star}^{(i)} + 1, & v \notin \mathcal{N}_C^{(i)} \cup \{\mathtt{c}_\perp^{(i)}\}, \end{cases} \quad > 0.$$

*Hence, the training data are linearly separable by a direction in* $\mathbb{R}_{\geq 0}^2$.

*Proof.* The result holds because $\lambda_{\mathtt{c}_\star}^{(i)} > 0$, $\lambda_v^{(i)} \leq 1$. $\qquad \square$

Define the *hard-margin* value of a unit direction $u \in \mathbb{S}^1 \cap \mathbb{R}_{\geq 0}^2$ (where $\mathbb{S}^1 = \{u \in \mathbb{R}^2 : \|u\|_2 = 1\}$) as

$$\gamma(u) := \min_i \min_{v \neq c_\star^{(i)}} \langle u, \Delta_{i,v} \rangle.$$

The corresponding *maximum-margin* direction is

$$u^\star \in \arg\max_{u \in \mathbb{S}^1 \cap \mathbb{R}_{\geq 0}^2} \gamma(u). \tag{18}$$

We characterize $u^\star$ using the following two quantities of the training sets:

$$\lambda_\star := \min_i \lambda_{c_\star}^{(i)} \in (0, 1], \qquad \Delta_{\text{train}} := \max_i \max_{v \in \mathcal{N}_C^{(i)} \setminus \{c_\star^{(i)}\}} \left(\lambda_v^{(i)} - \lambda_{c_\star}^{(i)}\right)_+ \in [0, 1],$$

where $(x)_+ := \max\{x, 0\}$. Intuitively, $\lambda_\star$ is the smallest mass ever placed on a reachable candidate across the training set, and $\Delta_{\text{train}}$ is the largest overshoot of a non-candidate but reachable node's weight relative to the reachable candidate.

**Lemma 10** (Closed-form lower envelope of the margin). *For any unit $u = (u_A, u_R) \in \mathbb{S}^1 \cap \mathbb{R}_{\geq 0}^2$,*

$$\gamma(u) = \min\{u_A \lambda_\star, \ u_R - u_A \Delta_{\text{train}}, \ u_A \lambda_\star + u_R\} = \min\{u_A \lambda_\star, \ u_R - u_A \Delta_{\text{train}}\}.$$

*Proof.* According to (17), we can directly obtain the the lower bounds $u_A \lambda_{c_\star}^{(i)}$, $u_R + u_A(\lambda_{c_\star}^{(i)} - \lambda_v^{(i)})$, and $u_A \lambda_{c_\star}^{(i)} +