# OpenReview forum: "Emergence of Superposition: Unveiling the Training Dynamics of Chain of Continuous Thought"
_ICLR.cc/2026/Conference — ICLR 2026 Poster_

### Official Review · Reviewer_dwvn · 2025-10-24

**Soundness:** 3
**Presentation:** 3
**Contribution:** 3
**Rating:** 8
**Confidence:** 3

**Summary:**

This paper focuses on the gap of whether superpositional reasoning (superposition) in continuous Chain-of-Thought (continuous CoT) naturally emerges under standard gradient training, chooses directed graph reachability, and provides a theoretical analysis of the training dynamics of a simplified two-layer Transformer, explicitly distinguishing the two phases of “thought generation” and “answer prediction” to explain the source of the mechanism; the core conclusion is that under continuous-thought training, the “index-matching logit” (which characterizes local search strength) first increases and then, under mild assumptions, remains bounded, thereby achieving an exploration–exploitation balance under uncertainty: it both leverages local structure to advance and assigns comparable weight to multiple candidate paths, so superposition arises naturally; the authors also validate this by experiments that track the growth of logits.

**Strengths:**

1. Focused problem that fills a gap: the paper centers on the core mechanism of why superposition would naturally arise during training in continuous Chain-of-Thought (continuous CoT) and provides an explanation from the perspective of training dynamics, going beyond prior work that only proves the existence of feasible solutions/expressivity without discussing whether they can be learned by gradient methods. The abstract and introduction clearly define the goal and differentiated positioning.

2. Solid key theoretical results: it formally compares two objectives (COCONUT vs. COCONUT-BFS), proving that under the former the index-matching logit is bounded whereas under the latter it diverges; it thus concludes that a bounded logit leads to a smoother next-step distribution, thereby retaining multiple candidate paths and promoting superposition. This conclusion offers a convincing explanation for why training does not over-commit to a single path.

3. Mutual support between empirics and theory: a two-layer GPT-2 decoder trained from scratch achieves 96.2% accuracy on a subset of ProsQA; during training the “frontier vs. non-frontier” logit gap first rises and then stabilizes, consistent with the “bounded” theory; length extrapolation appears at longer steps. These observations support the chain “bounded logit → superposition/exploration–exploitation balance.”

**Weaknesses:**

1. Ablation and robustness analyses are somewhat limited: there is a lack of systematic ablations over factors such as number of layers/heads/width and learning rate; nor are results under different random seed settings presented (there is only a verbal note in a footnote that different random seeds were used). The dataset is limited to a subset of ProsQA. It is recommended that the authors conduct more detailed experimental validation on more related datasets and under more settings.

2. (Small suggestion) To facilitate readers’ quick understanding of the paper’s contributions, it is recommended to describe an informal version of the conclusions in the first half of the paper.

**Questions:**

Based on your theoretical understanding, can you propose any targeted insights or approaches for enhancing a model’s reasoning ability during the training process?

---

> ### Author Response · Authors · 2025-11-21
>
> We thank the reviewer for the helpful and insightful feedback. Below are our responses.
>
> > Ablation and robustness analyses are somewhat limited: there is a lack of systematic ablations over factors such as number of layers/heads/width and learning rate; nor are results under different random seed settings presented (there is only a verbal note in a footnote that different random seeds were used). The dataset is limited to a subset of ProsQA. It is recommended that the authors conduct more detailed experimental validation on more related datasets and under more settings.
>
> We thank the reviewer for the suggestion. We added Section F.1 (page 29,31) and Table 4 (page 29) in the updated version, where we evaluated the sensitivity of COCONUT training to model depth, number of attention heads, hidden width, and learning rate. We include the accuracy number below.
>
> | # of layers | accuracy |
> |-------------|----------|
> | 2           | 98.8     |
> | 4           | 97.3     |
> | 8           | 96.5     |
> | 12          | 67.4     |
>
> | # of heads | accuracy |
> |------------|----------|
> | 4          | 98.0     |
> | 8          | 98.8     |
> | 12         | 98.8     |
>
> | width | accuracy |
> |-------|----------|
> | 384   | 62.0     |
> | 768   | 98.8     |
> | 1536  | 97.7     |
>
> | learning rate | accuracy |
> |----------------|----------|
> | 2e-4          | 58.1     |
> | 1e-4          | 98.8     |
> | 5e-5          | 62.1     |
>
> We observe that models with $L = 4, 8$ layers maintain high accuracy, while $L=12$ is
> harder to optimize. The performance remains comparable when the model width $d \in$ \{ 768, 1536 \}, but degrades when the width is too small (e.g., $d = 384$). Varying the number of heads does not have major effects on final accuracy, whereas too large or too small learning rates tend to degrade performance. Weight tying setting does not affect model performance.
>
> We emphasize that each ablation in the above tables varies only a single hyperparameter at a time, keeping all other settings identical to our main experiment. In practice, these hyperparameters interact in a coupled manner. For instance, with a smaller learning rate of 5e-5, we can extend the first-stage training to 300 epochs and get 97.0\% accuracy. For deeper models with $L = 12$, prolonging first-stage training to 400 epochs and reducing the learning rate to 5e-5 improves accuracy to 99.6\%. A comprehensive hyperparameter interaction study is beyond the scope of this work and is left for future investigation.
>
>  For random seeds, we ran 3 random seeds following the main experiment setting, and the accuracy was $97.9 \pm 1.5$.
>
> We also answered this question in CQ4 in the global response.
>
> >  To facilitate readers’ quick understanding of the paper’s contributions, it is recommended to describe an informal version of the conclusions in the first half of the paper.
>
> Thanks for the great suggestion!  We added Section 2.1 (from line 180 to line 233) to the updated version, which serves as an intuitive introduction and visualization of the superposition mechanism and the index-matching logit $\mu$ before delving into the technical details in Section 3.
>
>
> We also answered this question in  CQ1 in the global response.
>
>
> > Based on your theoretical understanding, can you propose any targeted insights or approaches for enhancing a model’s reasoning ability during the training process?
>
> We discuss two practical insights from our theory that could improve training recipes.
>
> First, according to our theoretical analysis and experiments, when different reasoning steps perform similar tasks (e.g., one-step expansion) using similar skills (e.g., local search), early-stage learning can help later-stage learning. This indicates that during curriculum learning, one does not necessarily need to allocate the same amount of resources to each stage; instead, one could allocate more compute to early stages to learn critical skills (such as the copy mechanism in the first layer in stage 1), and less compute to later stages to better utilize computational resources.
>
> Second, as our analysis shows that a positive, not-too-large index-matching logit can balance exploration and exploitation, one could apply logit clipping, especially when continuous CoT is used, to better prevent the model from overly confidently committing to wrong solutions.
>
> We also answered this question in CQ 5 in the global response.

---

### Official Review · Reviewer_eBsL · 2025-10-27

**Soundness:** 3
**Presentation:** 2
**Contribution:** 3
**Rating:** 6
**Confidence:** 4

**Summary:**

This paper offers a theoretical and empirical account of how the superposition principle arises in transformers under gradient-based training. Focusing on a two-layer transformer with continuous-thought decoding, the authors delineate two curriculum stages: (i) autoregressive generation of continuous thoughts and (ii) final prediction from the aggregated latent sequence. They show that, under an appropriate loss, attention logits remain bounded in Stage I, which expands the next thought. Leveraging max-margin analysis, they establish direction convergence and obtain accuracy guarantees on unseen examples.

**Strengths:**

- Theoretical rigor: Under some assumptions, the paper provides rigorous theory about how bounded attention logits and superposition naturally arise from gradient-based training. From a methodological point of view, this paper simplifies the difficulties of theoretical analysis by closing the dynamics through some assumptions.
- Insight into reasoning processes: Through the focus on the graph reachability problem, the analysis reveals concrete relationships between loss design and the evolution of attention weights.
- Experimental verification: The theoretical results were partially verified through some experiments. Section 5 provide the evidence that the theoretically predicted logit behaviors manifest during actual learning.

**Weaknesses:**

- The clarity of the article's presentation needs significant improvement. For example:

  - In the third subsection of section 3, the article introduces prior knowledge about the first decoder layer (Eq. (1)) based on some related work. This should be stated as an assumption and verified through experiments.
  - Similarly, the reparameterization introduced in Eq. (3) and Eq. (7) should also be considered as an assumption and partially verified through experiments.
  -  In section 5.1, the explanation of $\mu$ is confusing. More intuition or discussion is needed to explain why the chosen logit can be a proxy for $\mu$. In contrast, the explanation in 5.2 does not have this problem.
  - On page 5, the fourth line from the bottom considers two losses under permutation. Intuitively, the permutation invariance of the data will lead to similar properties of the parameters (as shown in Lem. 1), but the relevant insight needs to be pointed out more explicitly.
  - The statement of Theorem 2 needs to be improved. According to the expression of $\beta$, all parameters should be regarded as vectors, and the theorem should show the boundedness of the operator between vectors.

- The plausibility of the assumptions or motivations is unclear.
  - In Eq. (3), the initialization of $\mu$ is setted as 0. However, it is in an inductive process. Specifically, this is step c of the course learning. Assuming $\mu$ is 0 does not seem reasonable considering that the attention parameters have been changed during the previous training.  Additional discussion or proof is needed to complete the inductive process.
  - As mentioned before, priors about the behavior of reparameterization and the first layer of attention need to be empirically supported.

- The significance of some experiments is unclear.

In Section 5.2, the article analyzes the attention of the answer token and the reasoning token. Although it shows similar results to the first half of Theorem 3, it does not show how the behavior of accuracy changes as the direction converges.

- Scalability of experiments.
The paper experiments on a two-layer transformer. The relevant experiments, definitions, and detailed analysis are not yet clear in their guidance for training with more layers or for attention behavior. A supplementary example with more layers would be helpful.

**Questions:**

Please refer to the Weaknesses.

---

> ### Author Response · Authors · 2025-11-21
>
> We thank the reviewer for the helpful and insightful feedback. Below are our responses.
>
> > In the third subsection of section 3, the article introduces prior knowledge about the first decoder layer (Eq. (1)) based on some related work. This should be stated as an assumption and verified through experiments.
>
> We thank the reviewer for the suggestion. Yes, we agree that for theoretical analysis, this should be stated as an assumption. For experimental verification, previous work [1] already provided empirical evidence that the first layer of the transformer performs the copy mechanism (e.g., Figure 5 in [1]). Additionally, we added Figure 7 and Section F.2 on page 31 to our revised version, which shows that even for transformers with more than two layers, Eq. (1) still matches the learned mechanism in the first layer.
>
> > Similarly, the reparameterization introduced in Eq. (3) and Eq. (7) should also be considered as an assumption and partially verified through experiments.
>
> Thanks for your suggestion. For Eq. (3), we assume the “off-diagonal” terms are zero since the quantities that matter are the differences between “diagonal” and “off-diagonal” terms. It is partially verified by the experiments in Figure 3, which track the logit difference. For Eq. (7), we assumed that the final answer token attends to the reasoning token to obtain information about the last thought.  In experiments in Figure 6, we show several possible mechanisms for the final answer token to obtain the last thought’s information, which are functionally equivalent to each other. We will state more explicitly that the reparameterizations are assumptions for theoretical analysis.
>
> > In section 5.1, the explanation of  is confusing. More intuition or discussion is needed to explain why the chosen logit can be a proxy for . In contrast, the explanation in 5.2 does not have this problem.
>
> According to Eq. (3), $\mu$ represents the (scaled) logit of the current thought to an edge where its source node is contained in the superposition in the current thought. In section 5.1, we choose a subset of those edges, namely the frontier edges, since they receive the most attention during one-step expansion, and attention logits on other edges are negligible. Also, since in the reparameterization we assumed “off-diagonal” terms to be zero because we focus on the logit difference, in practice, we use the logit difference between the frontier and non-frontier edges as an approximation of $\mu$.
>
> > On page 5, the fourth line from the bottom considers two losses under permutation. Intuitively, the permutation invariance of the data will lead to similar properties of the parameters (as shown in Lem. 1), but the relevant insight needs to be pointed out more explicitly.
>
> Thanks for pointing this out. We will state the insight you mentioned more explicitly in the revision.
>
> > The statement of Theorem 2 needs to be improved. According to the expression of beta, all parameters should be regarded as vectors, and the theorem should show the boundedness of the operator between vectors.
>
> Although the whole $\beta$ is a vector, each $\beta\_v$ is a scalar for different nodes $v$. Also, the main message of Theorem 2 is not showing the boundness of $\beta\_v$, but the positivity of $\beta\_v$ for $v$ in $N\_{c+1}$ when the index-matching logit $\mu > 0$. We will state this more clearly.
>
> > In Eq. (3), the initialization of  is setted as 0. However, it is in an inductive process. Specifically, this is step c of the course learning. Assuming  is 0 does not seem reasonable considering that the attention parameters have been changed during the previous training. Additional discussion or proof is needed to complete the inductive process.
>
> We thank the reviewer for pointing this out. We assumed a zero initialization to simplify our theoretical analysis. However, the analysis can be directly extended to the case for a non-zero initialization. For example, in Theorem 1, the result for the convergence case does not rely on initialization; for the divergence case, $\mu(t)$ still diverges with at least a logarithmic rate. We will state this more clearly.
>
> > As mentioned before, priors about the behavior of reparameterization and the first layer of attention need to be empirically supported.
>
> Thank the reviewer for the suggestion. We added Figure 7 and Section F.2 on page 31 to our revised version, which supports the priors regarding the first-layer attention. Experiments in Figures 3 and 6 also partially support the reparameterization.

---

> > ### Author Response · Authors · 2025-11-21
> >
> > > In Section 5.2, the article analyzes the attention of the answer token and the reasoning token. Although it shows similar results to the first half of Theorem 3, it does not show how the behavior of accuracy changes as the direction converges.
> >
> >
> > We added an accuracy curve during the answer-prediction stage in Figure 9 on Page 32 in our revised version. The accuracy shows a rapid improvement corresponding to the learning of residual carryover and candidate lift signals. We kindly refer the reviewer to Appendix F3 (page 31) and Figure 9 (page 32) for more details.
> >
> >
> >
> >
> > > Scalability of experiments. The paper experiments on a two-layer transformer. The relevant experiments, definitions, and detailed analysis are not yet clear in their guidance for training with more layers or for attention behavior. A supplementary example with more layers would be helpful.
> >
> > We thank the reviewer for the suggestion. We conducted experiments on transformers beyond two layers to validate that the superposition mechanism is ubiquitous in Section F.2 in the updated version. Specifically, we train a four-layer transformer using continuous CoT and inspect its internal mechanism. The observed reasoning patterns can be summarized as follows:
> >
> > -  **First layer (induction head, see Figure 7 on page 31):** The first layer performs token-level copying, propagating node information into edge tokens \<e\>, consistent with the copy mechanism derived in previous theoretical analysis [1].
> >
> > - **Second layer and beyond (superposition, see Figure 8 on page 32):** From the second layer onward, the model aggregates over reachable nodes in a superpositional representation that enables parallel breadth-first exploration.
> >
> > We kindly refer reviewers to Appendix F.2 and Figures 7, 8 for more details.
> >
> > We also answered this question in CQ3 in the global response.
> >
> >
> > **References:**
> >
> > [1] Zhu H, Hao S, Hu Z, Jiao J, Russell S, Tian Y. Reasoning by Superposition: A Theoretical Perspective on Chain of Continuous Thought. arXiv preprint arXiv:2505.12514. 2025 May 18.

---

### Official Review · Reviewer_iW6r · 2025-10-29

**Soundness:** 3
**Presentation:** 2
**Contribution:** 3
**Rating:** 6
**Confidence:** 2

**Summary:**

This work theoretically studies how the superposition mechanism in continuous CoT is learned from gradient-based methods. It derives theoretical results by formulating the training dynamics of a two-layer transformer on the graph reachability problem in two training stages of thought generation and prediction. It also reveals the importance of the bounded index-matching logit and its connection to the tradeoff between exploration and exploitation.

**Strengths:**

1. This work addresses an important problem: as the previous work (Zhu et al., 2025) theoretically shows that the core superiority of continuous CoT lies in reason by superposition, this work solves a next question: how the superposition can be learned by gradient-based methods.

2. The contributions of this work are sufficient as it derives multiple novel theoretic results on the gradient-based learned superposition mechanism.

**Weaknesses:**

1. The presentation should be improved.
(1) The draft contains many typos and inconsistency, which should be thoroughly revised.
(2) As the content is mostly hard to follow, more intuitive visuals may need to be added for better understanding.

2. This work discusses only two-layer transformer. It would be better to discuss about the generalization to models with more layers.

3. Practical implication may be supplemented. Based on the theorems proposed in this work, does one design a better training recipe for continuous CoT?

**Questions:**

Please refer to the Weaknesses.

---

> ### Author Response · Authors · 2025-11-21
>
> We thank the reviewer for the helpful and insightful feedback. Below are our responses.
>
> > 1, The presentation should be improved. (1) The draft contains many typos and inconsistency, which should be thoroughly revised. (2) As the content is mostly hard to follow, more intuitive visuals may need to be added for better understanding.
>
> Thank the reviewer for the suggestion.  We added Section 2.1 (from line 180 to line 233) to the updated version, which serves as an intuitive introduction and visualization of the superposition mechanism and the index-matching logit $\mu$ before delving into the technical details in Section 3.
>
> We also answered this question in CQ1 in the global response.
>
> > 2. This work discusses only two-layer transformer. It would be better to discuss about the generalization to models with more layers.
>
> Thanks for your suggestions! We conducted experiments on transformers beyond two layers to validate that the superposition mechanism is ubiquitous in Section F.2 in the updated version. Specifically, we train a four-layer transformer using continuous CoT and inspect its internal mechanism. The observed reasoning patterns can be summarized as follows:
>
> - **First layer (induction head, see Figure 7 on page 31):** The first layer performs token-level copying, propagating node information into edge tokens \<e\>, consistent with the copy mechanism derived in previous theoretical analysis [1].
>
> - **Second layer and beyond (superposition, see Figure 8 on page 32):** From the second layer onward, the model aggregates over reachable nodes in a superpositional representation that enables parallel breadth-first exploration.
>
> We kindly refer reviewers to Appendix F.2 and Figures 7, 8 for more details.
>
> We also answered this question in CQ3 in the global response.
>
> > 3. Practical implication may be supplemented. Based on the theorems proposed in this work, does one design a better training recipe for continuous CoT?
>
> We discuss two practical insights from our theory that could improve training recipes.
>
> First, according to our theoretical analysis and experiments, when different reasoning steps perform similar tasks (e.g., one-step expansion) using similar skills (e.g., local search), early-stage learning can help later-stage learning. This indicates that during curriculum learning, one does not necessarily need to allocate the same amount of resources to each stage; instead, one could allocate more compute to early stages to learn critical skills (such as the copy mechanism in the first layer in stage 1), and less compute to later stages to better utilize computational resources.
>
> Second, as our analysis shows that a positive, not-too-large index-matching logit can balance exploration and exploitation, one could apply logit clipping, especially when continuous CoT is used, to better prevent the model from overly confidently committing to wrong solutions.
>
> We also answered this question in CQ5 in the global response.
>
> **References:**
>
> [1] Zhu H, Hao S, Hu Z, Jiao J, Russell S, Tian Y. Reasoning by Superposition: A Theoretical Perspective on Chain of Continuous Thought. arXiv preprint arXiv:2505.12514. 2025 May 18.

---

### Official Review · Reviewer_ohkE · 2025-11-01

**Soundness:** 3
**Presentation:** 2
**Contribution:** 2
**Rating:** 6
**Confidence:** 2

**Summary:**

Building on recent work in continuous Chain-of-Thought, the authors seek to explain why the training procedure yields strong reasoning behavior. They decompose the model’s prediction into two stages, develop a mathematical analysis focusing on the index-matching logit, an important quantity in the model, and present experiments that support their claims.

**Strengths:**

1. The paper formulates a clear, important problem in continuous Chain-of-Thought, explaining why such capability is enabled during training.
2. The claims are supported by precise mathematical derivations and well-designed experiments that substantiate the analysis.

**Weaknesses:**

1. The writing is careful but not intuitive. As a non-expert reader in this domain, it is important to provide a more intuitive background introduction. In this paper, it is especially important to introduce index-matching logic and related concepts gradually and intuitively.
2. The paper is focused on one single task of graph reachability. Could it be tested on more tasks and how would it be generalized to other tasks?

**Questions:**

See weaknesses.

---

> ### Author Response · Authors · 2025-11-21
>
> We thank the reviewer for the helpful and insightful feedback. Below are our responses.
>
> > 1. The writing is careful but not intuitive. As a non-expert reader in this domain, it is important to provide a more intuitive background introduction. In this paper, it is especially important to introduce index-matching logic and related concepts gradually and intuitively.
>
> Thank the reviewer for the suggestion! We added Section 2.1 (from line 180 to line 233) to the updated version, which serves as an intuitive introduction and visualization of the superposition mechanism and the index-matching logit $\mu$ before delving into the technical details in Section 3.
>
> We also answered this question in CQ1 in the global response.
>
> > 2. The paper is focused on one single task of graph reachability. Could it be tested on more tasks and how would it be generalized to other tasks?
>
> First, we argue that graph reachability is already a general task. **In theory**, graph reachability is NL(nondeterministic logarithmic-space)-complete, where many important theoretical problems can be reduced to it, such as the Turing machine halting problem. Therefore, the setting we studied in this paper can be extended to more general settings, such as Turing machines, where states can be modeled as nodes and transitions can be modeled as directed edges. **In practice**, the graph reachability problem abstracts many real-world reasoning problems (such as logic deduction, planning, and searching), where nodes represent entities (such as people, places, and countries) and directed edges capture relationships between entities. Moreover, it captures one of the most fundamental difficulties in many reasoning problems — identifying the correct path among numerous plausible candidates.
>
> Second, we focus on graph reachability because we aim to address the open question in prior work [1], which also studies graph reachability. Specifically, the previous work [1] uses graph reachability and provides a construction of the parameters, leaving the theoretical analysis of whether the construction can be naturally learned as future work; this paper fills the gap by theoretically analyzing how the gradient-based method naturally learns the superposition mechanism for graph reachability.
>
> We also agree with the reviewers that studying the mechanism of continuous CoT on tasks beyond graph reachability is very important. We acknowledge that our work primarily focuses on graph reachability and leave the study of tasks beyond it for future work.
>
> We also answered this question in CQ2 in the global response on task generality.
>
> **References:**
>
> [1] Zhu H, Hao S, Hu Z, Jiao J, Russell S, Tian Y. Reasoning by Superposition: A Theoretical Perspective on Chain of Continuous Thought. arXiv preprint arXiv:2505.12514. 2025 May 18.

---

### Official Review · Reviewer_Aa1k · 2025-11-11

**Soundness:** 3
**Presentation:** 3
**Contribution:** 3
**Rating:** 6
**Confidence:** 3

**Summary:**

This paper investigates the superposition mechanism underlying continuous chain-of-thought (continuous CoT, or COCONUT) in large language models (LLMs), a paradigm where reasoning traces are maintained in a continuous latent space rather than discrete tokens. Building on prior work that theoretically showed a two-layer transformer with continuous CoT can solve directed graph reachability via superposition of reasoning traces, the authors address a critical open question: Do gradient-based training methods naturally induce this superposition?

To answer this, the authors analyze the training dynamics of a simplified two-layer transformer on the directed graph reachability task, splitting training into two stages: (1) a thought-generation stage, where the model autoregressively expands continuous thoughts; and (2) a prediction stage, where it converts these thoughts into a final answer. Their core theoretical insight is that the index-matching logit—a metric of the model’s local search ability—remains bounded under mild assumptions during COCONUT training. This bounded logit balances exploration (assigning comparable weights to multiple plausible reasoning traces) and exploitation (leveraging local graph structure), directly enabling superposition. The authors validate these claims with experiments on a GPT-2-style model, showing logits stabilize at bounded values (consistent with theory) and the model achieves 96.2% test accuracy on graph reachability.

**Strengths:**

1. Mechanistic insight: The paper demystifies a core advantage of continuous CoT—superposition—by tying it to a measurable quantity (index-matching logit) and its training dynamics. This is a rare example of "reverse-engineering" why a latent reasoning method works, rather than just showing it works.

2. Theory-experiment alignment: Experiments are not just "proof of concept" but directly validate theoretical predictions (e.g., logit saturation, length generalization to c=3,4 despite training only on c=1,2). This strengthens confidence in the mechanistic claims.

3. Practical relevance: The focus on single demonstrations per sample (COCONUT) reflects real-world training constraints (where large CoT datasets are costly to annotate). This makes the results more applicable than methods relying on exhaustive BFS-style supervision.

**Weaknesses:**

1. Task generality: The analysis is limited to directed graph reachability, a synthetic task with well-defined "reasoning traces" (paths). It is unclear if the superposition mechanism extends to tasks with more ambiguous traces (e.g., open-domain QA, where reasoning steps are less structured). The paper should acknowledge this limitation and suggest future work on diverse tasks.

2. Model scalability: The experiments use a small two-layer model. While this is standard for mechanistic work, the paper could include a brief discussion of how deeper models might interact with superposition (e.g., whether layers stack superposition or refine it) to address scalability concerns.

3. Alternative mechanisms: The paper does not rule out competing explanations for superposition. For example, could the model’s weight tying (assumed for convenience) or curriculum learning (multi-stage training) also contribute to superposition? A short ablation on weight tying or curriculum learning would strengthen the causal link between COCONUT training and superposition.

**Questions:**

1. The paper notes that logits plateau in practice but grow unbounded in theory, attributed to interactions between thought-generation and prediction stages. Could you elaborate on why these interactions cause plateauing? For example, does the prediction stage’s loss implicitly regularize the thought-generation logits, or is it a numerical effect (e.g., gradient clipping in practice)?

2. The model exhibits "length generalization" (e.g., generating c=4 continuous thoughts despite training on c≤2). Do you have evidence that this generalization relies on superposition? For example, if you perturb the superposition (e.g., zero out weights for non-optimal traces), does length generalization break?

3. The theoretical analysis assumes linear attention. How might non-linear attention (e.g., softmax attention, the standard in LLMs) affect the index-matching logit’s dynamics? Would softmax attention’s normalization further bound logits, or introduce new complexities (e.g., vanishing gradients)?

---

> ### Author Response · Authors · 2025-11-21
>
> We thank the reviewer for the helpful and insightful feedback. Below are our responses.
>
> > 1. Task generality: The analysis is limited to directed graph reachability ... The paper should acknowledge this limitation and suggest future work on diverse tasks.
>
> First, we argue that graph reachability is already a general task. **In theory**, graph reachability is NL(nondeterministic logarithmic-space)-complete, where many important theoretical problems can be reduced to it, such as the Turing machine halting problem. Therefore, the setting we studied in this paper can be extended to more general settings, such as Turing machines, where states can be modeled as nodes and transitions can be modeled as directed edges. **In practice**, the graph reachability problem abstracts many real-world reasoning problems (such as logic deduction, planning, and searching), where nodes represent entities (such as people, places, and countries) and directed edges capture relationships between entities. Moreover, it captures one of the most fundamental difficulties in many reasoning problems — identifying the correct path among numerous plausible candidates.
>
> Second, we focus on graph reachability because we aim to address the open question in prior work [1], which also studies graph reachability. Specifically, the previous work [1] uses graph reachability and provides a construction of the parameters, leaving the theoretical analysis of whether the construction can be naturally learned as future work; this paper fills the gap by theoretically analyzing how the gradient-based method naturally learns the superposition mechanism for graph reachability.
>
> We also agree with the reviewers that studying the mechanism of continuous CoT on tasks beyond graph reachability is very important. We acknowledge that our work primarily focuses on graph reachability and leave the study of tasks beyond it for future work.
>
> We also answered this question in CQ2 in the global response.
>
> > 2. Model scalability: The experiments use a small two-layer model ...
>
> Thanks for your suggestions! We conducted experiments on transformers beyond two layers to validate that the superposition mechanism is ubiquitous in Section F.2 in the updated version. Specifically, we train a four-layer transformer using continuous CoT and inspect its internal mechanism. The observed reasoning patterns can be summarized as follows:
>
> - **First layer (induction head, see Figure 7 on page 31):** The first layer performs token-level copying, propagating node information into edge tokens \<e\>, consistent with the copy mechanism derived in previous theoretical analysis [1].
>
> - **Second layer and beyond (superposition, see Figure 8 on page 32):** From the second layer onward, the model aggregates over reachable nodes in a superpositional representation that enables parallel breadth-first exploration.
>
> We kindly refer reviewers to Appendix F.2 and Figures 7, 8 for more details.
>
> We also answered this question in CQ3 in the global response.
>
> > 3. Alternative mechanisms: The paper does not rule out competing explanations for superposition ...
>
> We thank the reviewer for the suggestion. For the weight tying, we added ablation experiments in Section F.1 (pages 29 and 31) and Table 4 (page 29), where both tied and untied weights achieve 98.8% accuracy, indicating that weight tying did not contribute to superposition. For curriculum learning, our experiments in Section 5.1 show that training only on stages 1 and 2 can learn the desired pattern for later stages, indicating that the superposition mechanism does not necessarily rely on curriculum learning.

---

> > ### Author Response · Authors · 2025-11-21
> >
> > > 1. The paper notes that logits plateau in practice but grow unbounded in theory, attributed to interactions between thought-generation and prediction stages. Could you elaborate on why these interactions cause plateauing? For example, does the prediction stage’s loss implicitly regularize the thought-generation logits, or is it a numerical effect (e.g., gradient clipping in practice)?
> >
> > A possible reason is that in theoretical analysis, we assume the weights of the superposition of the last thought are fixed, which makes the feature for the final prediction clean and separable. However, in experiments, since training on the prediction stage changes the model weights, which then changes the weights of the superposition of the last thought (note that thought generation and final prediction use the same model), the features for the final prediction might not be clearly separable, and thus make the logit for final prediction plateau.
> >
> > > 2. The model exhibits "length generalization" (e.g., generating c=4 continuous thoughts despite training on c≤2). Do you have evidence that this generalization relies on superposition? For example, if you perturb the superposition (e.g., zero out weights for non-optimal traces), does length generalization break?
> >
> > The “length generalization” does not rely on superposition. In fact, the length generalization pattern is quantified by the index-matching logit $\mu$, which quantifies the model’s local search capability. This capability is determined by the model weights learned from training, which are independent of the superposition during inference. In other words, zeroing out weights for non-optimal traces only changes the states during inference but does not change the model weights, and thus does not break the length generalization capability quantified by the index-matching logit.
> >
> > > 3. The theoretical analysis assumes linear attention. How might non-linear attention (e.g., softmax attention, the standard in LLMs) affect the index-matching logit’s dynamics? Would softmax attention’s normalization further bound logits, or introduce new complexities (e.g., vanishing gradients)?
> >
> > The softmax attention mainly normalizes the distribution, but also magnifies the difference between the weights of two nodes when the index-matching logit is large. For example, when the index-matching logit $\mu = 10$, one node has indegree 2 and another has indegree 1, linear attention results in weights 20 vs. 10, while softmax attention results in an exp(20): exp(10) ratio of weights, which is exponentially large. We use linear attention for simplicity in our theoretical analysis, but the main message remains the same as with softmax attention.
> >
> > **References:**
> >
> > [1] Zhu H, Hao S, Hu Z, Jiao J, Russell S, Tian Y. Reasoning by Superposition: A Theoretical Perspective on Chain of Continuous Thought. arXiv preprint arXiv:2505.12514. 2025 May 18.

---

### Author Response · Authors · 2025-11-21
**Global Response (1)**

We thank reviewers for the time and effort in reviewing our paper and providing insightful and helpful feedback. Below, we first address some common questions (CQ).

### CQ 1: Adding intuitive introductions and visualization of the mechanism

Answer:

We thank the reviewers for the great suggestions. We added Section 2.1 (from line 180 to line 233) to the updated version, which serves as an intuitive introduction and visualization of the superposition mechanism and the index-matching logit $\mu$ before delving into the technical details in Section 3.


### CQ 2: Task generality

Answer:

First, we argue that graph reachability is already a general task. **In theory**, graph reachability is NL(nondeterministic logarithmic-space)-complete, where many important theoretical problems can be reduced to it, such as the Turing machine halting problem. Therefore, the setting we studied in this paper can be extended to more general settings, such as Turing machines, where states can be modeled as nodes and transitions can be modeled as directed edges. **In practice**, the graph reachability problem abstracts many real-world reasoning problems (such as logic deduction, planning, and searching), where nodes represent entities (such as people, places, and countries) and directed edges capture relationships between entities. Moreover, it captures one of the most fundamental difficulties in many reasoning problems — identifying the correct path among numerous plausible candidates.

Second, we focus on graph reachability because we aim to address the open question in prior work [1], which also studies graph reachability. Specifically, the previous work [1] uses graph reachability and provides a construction of the parameters, leaving the theoretical analysis of whether the construction can be naturally learned as future work; this paper fills the gap by theoretically analyzing how the gradient-based method naturally learns the superposition mechanism for graph reachability.

We also agree with the reviewers that studying the mechanism of continuous CoT on tasks beyond graph reachability is very important. We acknowledge that our work primarily focuses on graph reachability and leave the study of tasks beyond it for future work.




### CQ 3: Mechanism of multi-layer transformers (more than two layers)

Answer:

We conducted experiments on transformers beyond two layers to validate that the superposition mechanism is ubiquitous in Section F.2 in the updated version. Specifically, we train a four-layer transformer using continuous CoT and inspect its internal mechanism. The observed reasoning patterns can be summarized as follows:

  - **First layer (induction head, see Figure 7 on page 31):** The first layer performs token-level copying, propagating node information into edge tokens \<e\>, consistent with the copy mechanism derived in previous theoretical analysis [1].

-  **Second layer and beyond (superposition, see Figure 8 on page 32):** From the second layer onward, the model aggregates over reachable nodes in a superpositional representation that enables parallel breadth-first exploration.

We kindly refer reviewers to Appendix F.2 and Figures 7, 8 for more details.

---

> ### Author Response · Authors · 2025-11-21
> **Global Response (2)**
>
> ### CQ 4: Model scalability and robustness against architectural and optimization choices
>
> Answer:
>
> We added Section F.1 (page 29,31) and Table 4 (page 29) in the updated version, where we evaluated the sensitivity of COCONUT training to model depth, number of attention heads, hidden width, and learning rate. We include the accuracy number below.
>
> | # of layers | accuracy |
> |-------------|----------|
> | 2           | 98.8     |
> | 4           | 97.3     |
> | 8           | 96.5     |
> | 12          | 67.4     |
>
> | # of heads | accuracy |
> |------------|----------|
> | 4          | 98.0     |
> | 8          | 98.8     |
> | 12         | 98.8     |
>
> | width | accuracy |
> |-------|----------|
> | 384   | 62.0     |
> | 768   | 98.8     |
> | 1536  | 97.7     |
>
> | learning rate | accuracy |
> |----------------|----------|
> | 2e-4          | 58.1     |
> | 1e-4          | 98.8     |
> | 5e-5          | 62.1     |
>
> We observe that models with $L = 4, 8$ layers maintain high accuracy, while $L=12$ is
> harder to optimize. The performance remains comparable when the model width $d \in$ \{ 768, 1536 \}, but degrades when the width is too small (e.g., $d = 384$). Varying the number of heads does not have major effects on final accuracy, whereas too large or too small learning rates tend to degrade performance. Weight tying setting does not affect model performance.
>
> We emphasize that each ablation in the above tables varies only a single hyperparameter at a time, keeping all other settings identical to our main experiment. In practice, these hyperparameters interact in a coupled manner. For instance, with a smaller learning rate of 5e-5, we can extend the first-stage training to 300 epochs and get 97.0\% accuracy. For deeper models with $L = 12$, prolonging first-stage training to 400 epochs and reducing the learning rate to 5e-5 improves accuracy to 99.6\%. A comprehensive hyperparameter interaction study is beyond the scope of this work and is left for future investigation.
>
>
> ### CQ 5: Practical insights from our theory
>
> Answer:
>
> We discuss two practical insights from our theory that could improve training recipes.
>
> First, according to our theoretical analysis and experiments, when different reasoning steps perform similar tasks (e.g., one-step expansion) using similar skills (e.g., local search), early-stage learning can help later-stage learning. This indicates that during curriculum learning, one does not necessarily need to allocate the same amount of resources to each stage; instead, one could allocate more compute to early stages to learn critical skills (such as the copy mechanism in the first layer in stage 1), and less compute to later stages to better utilize computational resources.
>
> Second, as our analysis shows that a positive, not-too-large index-matching logit can balance exploration and exploitation, one could apply logit clipping, especially when continuous CoT is used, to better prevent the model from overly confidently committing to wrong solutions.
>
> **References:**
>
> [1] Zhu H, Hao S, Hu Z, Jiao J, Russell S, Tian Y. Reasoning by Superposition: A Theoretical Perspective on Chain of Continuous Thought. arXiv preprint arXiv:2505.12514. 2025 May 18.

---

### Meta-Review · Area_Chair_L2Md · 2026-01-10

**Summary:**

This paper theoretically investigates the training dynamics of a simplified two-layer Transformer on directed graph reachability problems, specifically focusing on the Chain of Continuous Thought (continuous CoT) framework. The authors unveil how a superposition mechanism, which allows the model to maintain and explore multiple reasoning traces simultaneously, naturally emerges from gradient-based training. The analysis breaks the training process into two distinct stages: (i) thought-generation and (ii) prediction. A key finding is the behavior of the "index-matching logit," which reflects the model's local search ability and balances exploration and exploitation to result in superposition.

The paper provides a compelling explanation for how complex reasoning mechanisms like superposition are learned via gradient descent (GD). By demonstrating that the model can exploit local structures to identify plausible search traces while maintaining comparable weights across them when uncertain, the work offers a rigorous insight into the LLM reasoning dynamics.

The pros and cons of this paper can be summarized as follows:
- **Pros:**
(Theoretical novelty) The theoretical results are both rigorous and solid. The identification of the superposition mechanism as a bridge between implicit parallel thinking and continuous CoT is a highly interesting and significant perspective.
(Numerical experiments) The authors provide experimental results tracking the growth of logits during training, which further validate their theoretical analysis regarding bounded index-matching logits and the emergence of superposition.

- **Cons:**
(Limited problem setting) While the directed graph reachability problem is fundamental and underlies many complex reasoning tasks, the current scope is somewhat limited. Extending the analysis to more diverse or higher-order reasoning problems would strengthen the generalizability of the findings.
(Writing) The reviewers noted several typos and suggested that some of the more dense theoretical sections could benefit from more intuitive, high-level explanations to improve accessibility for a broader audience.


In summary, despite the relatively narrow task scope and the need for minor writing improvements, this paper presents a rigorous, solid, and novel theoretical contribution. It provides a new insight on the training dynamics of LLMs on logical reasoning tasks so that it can be accepted at ICLR.

**Reviewer Concerns:**

I think there is no particular concern on reviewers.

**Reviewer Scores:**

Overall the review scores are already sufficiently high. I think the reviewers would not have raised the scores further more even after the discussions.

---

### Decision · Program_Chairs · 2026-01-26

Accept (Poster)